# Systematic analysis of exonic germline and postzygotic de novo mutations in bipolar disorder

Masaki Nishioka[1], An-a Kazuno[1], Takumi Nakamura[1,2], Naomi Sakai[1], Takashi Hayama[3], Kumiko Fujii[4], Koji Matsuo[5], Atsuko Komori[1], Mizuho Ishiwata[1], Yoshinori Watanabe[6], Takashi Oka[7], Nana Matoba [1,13], Muneko Kataoka[1,14], Ahmed N. Alkanaq[8], Kohei Hamanaka[8], Takashi Tsuboi[9], Toru Sengoku[10], Kazuhiro Ogata[10], Nakao Iwata [11], Masashi Ikeda [11], Naomichi Matsumoto [8], Tadafumi Kato [1,2✉] & Atsushi Takata [1,8,12✉]

Bipolar disorder is a severe mental illness characterized by recurrent manic and depressive episodes. To better understand its genetic architecture, we analyze ultra-rare de novo mutations in 354 trios with bipolar disorder. For germline de novo mutations, we find significant enrichment of loss-of-function mutations in constrained genes (corrected-$P$ = 0.0410) and deleterious mutations in presynaptic active zone genes (FDR = 0.0415). An analysis integrating single-cell RNA-sequencing data identifies a subset of excitatory neurons preferentially expressing the genes hit by deleterious mutations, which are also characterized by high expression of developmental disorder genes. In the analysis of postzygotic mutations, we observe significant enrichment of deleterious ones in developmental disorder genes ($P$ = 0.00135), including the *SRCAP* gene mutated in two unrelated probands. These data collectively indicate the contributions of both germline and postzygotic mutations to the risk of bipolar disorder, supporting the hypothesis that postzygotic mutations of developmental disorder genes may contribute to bipolar disorder.

[1] Laboratory for Molecular Dynamics of Mental Disorders, RIKEN Center for Brain Science, Wako, Saitama, Japan. [2] Department of Psychiatry and Behavioral Science, Juntendo University Graduate School of Medicine, Tokyo, Japan. [3] Yokohama Mental Clinic Totsuka, Yokohama, Japan. [4] Department of Psychiatry, Shiga University of Medical Science, Otsu, Shiga, Japan. [5] Department of Psychiatry, Saitama Medical University, Moroyama, Saitama, Japan. [6] Ichigaya Himorogi Clinic, Tokyo, Japan. [7] Juzen Hospital, Kanazawa, Ishikawa, Japan. [8] Department of Human Genetics, Yokohama City University Graduate School of Medicine, Yokohama, Japan. [9] Department of Life Sciences, Graduate School of Arts and Sciences, The University of Tokyo, Tokyo, Japan. [10] Department of Biochemistry, Yokohama City University Graduate School of Medicine, Yokohama, Japan. [11] Department of Psychiatry, Fujita Health University, Toyoake, Aichi, Japan. [12] Laboratory for Molecular Pathology of Psychiatric Disorders, RIKEN Center for Brain Science, Wako, Saitama, Japan. [13] Present address: Department of Genetics and UNC Neuroscience Center, The University of North Carolina at Chapel Hill, Chapel Hill, NC, USA. [14] Present address: Ebara Hospital, Tokyo Metropolitan Health and Hospitals Corporation, Tokyo, Japan. ✉email: tadafumi.kato@juntendo.ac.jp; atsushi.takata@riken.jp

Bipolar disorder (BD) is a common and severe neuropsychiatric disorder afflicting the patients and their families with depressive/manic episodes. The depressive episodes and psychotic symptoms are one of the global medical issues in our time[1]. Although there are some available ameliorative medications, these were serendipitously discovered or originally developed for other diseases, and the fundamental biological basis of BD is unknown.

Genetic and epidemiological studies have consistently demonstrated that BD is a highly heritable phenotype[2]. Therefore, numerous genetic studies for BD have been conducted. Though there had been considerable between-study inconsistencies until recently, large-scale genome-wide association studies (GWAS) of common single-nucleotide polymorphisms (SNPs) in this decade have revealed several tens of genetic loci robustly associated with BD and have contributed to a better understanding of the genetic architecture of BD[3,4]. On the other hand, the role of rare single-nucleotide variants including de novo mutations (DNMs) has currently been investigated only in studies in which small-to-moderate numbers of individuals were sequenced[5–9], while rare protein-truncating variants are reported to be enriched in BD[10]. Given the prominent success in studies of exonic DNMs in other neuropsychiatric disorders such as schizophrenia[11–16] and autism spectrum disorder (ASD)[17–20]—identifying many disease-causative genes and deepening our understanding of their disease etiology—it is worth conducting a larger study of DNMs in BD.

We, in this study, perform a comprehensive analysis of exonic DNMs in the protein-coding regions as rare variants potentially related to BD. We not only initially focused on germline DNMs (gDNMs) but also systematically analyze postzygotic (i.e., somatic) DNMs (pzDNMs) later. An overview of the study design is illustrated in Supplementary Fig. 1. First, we analyze 257 BD and 1640 control trio exome using a unified pipeline and evaluate overall patterns of gDNM enrichment in BD. Second, we construct a more comprehensive list of gDNMs in 354 BD trios and analyze the properties of the genes hit by deleterious gDNMs.

Third, we investigate genes recurrently hit by deleterious gDNMs in BD or a broad spectrum of neuropsychiatric/developmental disorders (DDs), including BD. Fourth, based on an observation of surely gene-disruptive pzDNM in a known neurodevelopmental disorder gene in BD, we perform a systematic survey of pzDNMs in BD. Overall, our results support the roles of both exonic gDNMs and pzDNMs in BD. Also, our analysis of the genes hit by deleterious gDNMs or pzDNMs in BD provides insights into its neurobiology, including biological pathways related to BD and neuronal cell types possibly playing a critical role in the disease etiology.

## Results

**Patterns of gDNM enrichment in BD.** To evaluate overall profiles of exonic gDNMs (all DNM variant calls identified in exons or splice sites by a standard pipeline for gDNMs; possibly including a small number of pzDNMs with a high variant allele fraction (VAF)) in BD, we analyzed exome data of 257 BD (115 bipolar I [BDI], 52 bipolar II [BDII], 2 BD not otherwise specified [BD-NOS], and 88 schizoaffective disorder [SCZAD] cases) and 1640 control trios (unaffected siblings of ASD probands) using a unified high-specificity analytical pipeline ("Methods"). As a previous DNM study for BD[9], we included SCZAD based on the phenotypic similarity[21], the genetic relationship[22–24], the shared drug response[25], and the suggested shared physiological traits[26] between SCZAD and BD. When all gDNMs, including those observed in the general population in the Genome Aggregation Database (gnomAD[27]; the non-neuro population) or the Tohoku Medical Megabank Organization (ToMMo[28]), were subjected to the analysis, there was no significant difference in the rates of any functional types (i.e., loss-of-function [LoF], damaging missense/inframe indel, non-damaging missense, and synonymous variants; see "Methods" and the legend of Fig. 1) of gDNMs between BD and controls (Fig. 1a). On the other hand, when we filtered out all the variants observed in gnomAD or ToMMo ("Methods"), we observed a trend toward the enrichment of LoF and

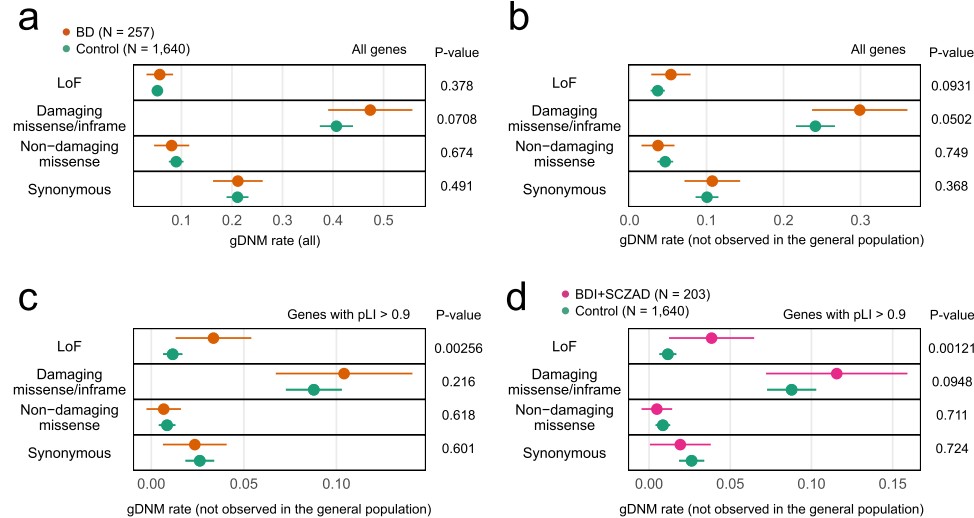

**Fig. 1 Patterns of gDNM enrichment in BD. a–d** Plots of per-individual rates of the following four types of gDNMs in the affected and unaffected groups: loss of function (LoF), damaging missense/inframe indel, non-damaging missense, and synonymous. Damaging missense gDNMs are defined as those with a Combined Annotation Dependent Depletion (CADD) score >15. **a** Rates of all gDNMs in BD (orange, N = 257) and controls (green, N = 1640). **b** Rates of gDNMs not observed in the general population (gnomAD and ToMMo) in BD and controls. **c** Rates of gDNMs not observed in the general population and hitting a constrained gene (pLI > 0.9) in BD and controls. **d** Rates of gDNMs not observed in the general population and hitting a constrained gene (pLI > 0.9) in bipolar I or schizoaffective disorder (BDI + SCZAD, magenta, N = 203) and controls. The mean of gDNM counts in the affected and unaffected groups for each mutational type is indicated as the colored points accompanied by the error bars (95% confidence intervals). Uncorrected P values calculated by one-tailed permutation tests are shown on the right of the plots.

damaging missense/inframe indel gDNMs in BD (Fig. 1b, uncorrected $P = 0.0931$ for LoF and 0.0502 for damaging missense/inframe indel), whereas the rates of non-damaging missense and synonymous gDNMs were similar between BD and controls. This pattern of enrichment is similar to the observations in studies of ASD and schizophrenia[15,29]. Based on this, we included gDNMs not found in gnomAD or ToMMo in the following analyses. We subsequently performed an analysis focusing on genes depleted for LoF variants in the general population (genes with a probability of being LoF-intolerant [pLI] score >0.90 in non-psychiatric population; we refer to these as "constrained" genes). We observed that LoF gDNMs in constrained genes are significantly enriched in BD (Fig. 1c, uncorrected $P = 0.00256$, corrected $P = 0.0410$; the Bonferroni method with the number of all tests in Fig. 1 [$n$ of tests $= 16$]). Besides, by excluding 54 trios with BDII or BD-NOS, we observed further prominent enrichment of LoF gDNMs (Fig. 1d, corrected $P = 0.0194$ [$n$ of tests $= 16$]) in constrained genes. Therefore, LoF gDNMs in constrained genes are particularly enriched in BD subtypes generally thought to be severe. We also found that the enrichment of LoF gDNMs in constrained genes remains significant when SCZAD individuals were excluded from the case group ($N$ of cases $= 169$, Supplementary Fig. 2), confirming that the observed enrichment is not solely explained by the SCZAD cases.

**Gene set enrichment analysis of deleterious gDNMs in BD.** Next, we analyzed the properties of the genes hit by gDNMs in BD. For this purpose, we compiled a more comprehensive list of gDNMs by (i) analyzing the exome data with multiple pipelines to achieve high sensitivity and (ii) utilizing the list of gDNMs in a published whole-genome sequencing study of BD[8] ("Methods"). After excluding 155 gDNMs that are observed in the general population, there are a total of 241 gDNMs in 354 BD trios (Supplementary Data 1). The per-individual rate of gDNMs including those observed in the general population (1.12) is consistent with previous studies. From the full list of gDNMs and their target genes, we subjected LoF or damaging missense/inframe indel (we refer to these as "deleterious") gDNMs in all genes to our gene set enrichment analysis ($n$ of applicable gDNMs $= 171$) to obtain insights into the biology of BD, based on our observation of nominal enrichment of deleterious gDNMs in BD (uncorrected $P = 0.0234$, comparison of deleterious gDNMs in BD [$N = 257$] and controls [$N = 1640$]), and considering the statistical power depending on the number of genes included in the analysis.

An unbiased and systematic gene ontology (GO) enrichment analysis of genes hit by deleterious gDNMs in BD by DNENRICH[13], which considers confounding factors such as gene sizes and local sequence contexts ("Methods"), identified four GO terms with a false discovery rate (FDR) <0.1 (Fig. 2a, top): presynaptic active zone (GO:0048786, FDR-corrected $P = 0.0415$), response to growth factor (GO:0070848, FDR-corrected $P = 0.0850$), neurotransmitter secretion (GO:0007269, FDR-corrected $P = 0.0850$), and divalent metal ion transport (GO:0070838, FDR-corrected $P = 0.0850$). We confirmed that these four terms are not enriched among the genes hit by deleterious gDNMs in controls compiled from ref. [20] (Fig. 2a, bottom). By visualizing networks of the 60 GO terms that showed nominal significance in BD (uncorrected $P < 0.05$, Supplementary Data 2) based on the similarity of the genes contained in each term, we observed the formation of six GO clusters, each of which is related to synapse, calcium ion, response to growth factor, regulation of metabolic processes, protein targeting to mitochondrion and endoplasmic reticulum, and kinase activity (Fig. 2b).

The genes hit by deleterious gDNMs in BD are enriched for constrained genes ($P = 0.00549$, DNENRICH analysis), while constrained genes are known to be enriched in synaptic genes[30]. Thus, the enrichment of synaptic genes including ion channel genes in the genes hit by deleterious gDNMs in BD would be reasonable.

We also performed a gene set enrichment analysis of deleterious gDNMs in BD integrating transcriptome datasets. A DNENRICH analysis of various human tissues in the Genotype-Tissue Expression (GTEx) project[31] (version 8) demonstrated that the genes hit by deleterious gDNMs in BD are most enriched for genes highly expressed in the anterior cingulate cortex (ACC) Brodmann Area 24 (Fig. 2c, uncorrected $P = 0.0129$), followed by kidney cortex (uncorrected $P = 0.0267$), hypothalamus (uncorrected $P = 0.0291$), and amygdala (uncorrected $P = 0.0302$). While there was no single tissue achieving the statistical significance after Bonferroni correction (Fig. 2c, $n$ of tests $= 54$), the genes with deleterious gDNMs in BD are more enriched for genes highly expressed in the brain when compared with non-brain tissues ($P = 1.08 \times 10^{-6}$, exact Wilcoxon rank-sum test). In an analysis utilizing the spatiotemporal transcriptome dataset of the human brains in BrainSpan[32], we observed that genes highly expressed in the late mid-fetal cortex are most enriched among the genes hit by deleterious gDNMs in BD (Fig. 2d, uncorrected $P = 0.00558$). Enrichment among the genes hit by deleterious gDNMs in BD was biased toward fetal periods when compared with postnatal developmental stages ($P = 9.78 \times 10^{-5}$, exact Wilcoxon rank-sum test). On the other hand, again there was no specific developmental period of brain region remained significant after Bonferroni correction ($n$ of tests $= 60$).

**Single-cell enrichment analysis of deleterious gDNMs in BD.** The above analysis integrating transcriptome data of bulk tissues showed an overall trend that the genes hit by deleterious gDNMs in BD are enriched for brain-expressed genes with a prenatal bias; however, we could not specify single brain regions with robust significance. By assuming that more sound and detailed insights can be obtained by performing an analysis being informed by transcriptome data with a higher, cellular-level resolution, we performed an integrative analysis of the genes hit by deleterious gDNMs in BD utilizing single-cell (nucleus) RNA sequencing data of adult human ACC, the brain region that showed the highest enrichment in Fig. 2c and have been consistently implicated in mood disorders[33,34].

When we visualized cell clusters in human ACC using uniform manifold approximation and projection (UMAP), we found a total of 16 clusters that are annotated by marker gene expression (Fig. 3a, b and Supplementary Fig. 3). We then identified the top 5% of the cells preferentially expressing the genes hit by deleterious gDNMs in BD using AUCell[35] and plotted them onto the UMAP embedding (the blue dots in Fig. 3c). We found that the majority of these cells fell into the clusters of excitatory neurons expressing *SLC17A7* (VGLUT1) (Fig. 3a–c). Among the six automatically detected clusters of excitatory neurons (clusters 0, 1, 3, 8, 10, and 12 in Fig. 3a), cells preferentially expressing the genes hit by deleterious gDNMs were especially enriched in the cluster 8 (Fig. 3c, d, $P = 6.78 \times 10^{-18}$ when compared with the theoretical expectation, one-tailed binomial test). By performing an analysis of genes differentially expressed between the cluster 8 cells and the other cells in the same large cluster of excitatory neurons (i.e., clusters 0, 1, 3, and 12), we found a total of 174 genes significantly upregulated (FDR < 0.05) in the cluster 8 cells (Supplementary Data 3, we refer to these genes as "c8 signature genes"). We confirmed that the genes hit by deleterious gDNMs in BD are significantly enriched for the c8 signature genes by a

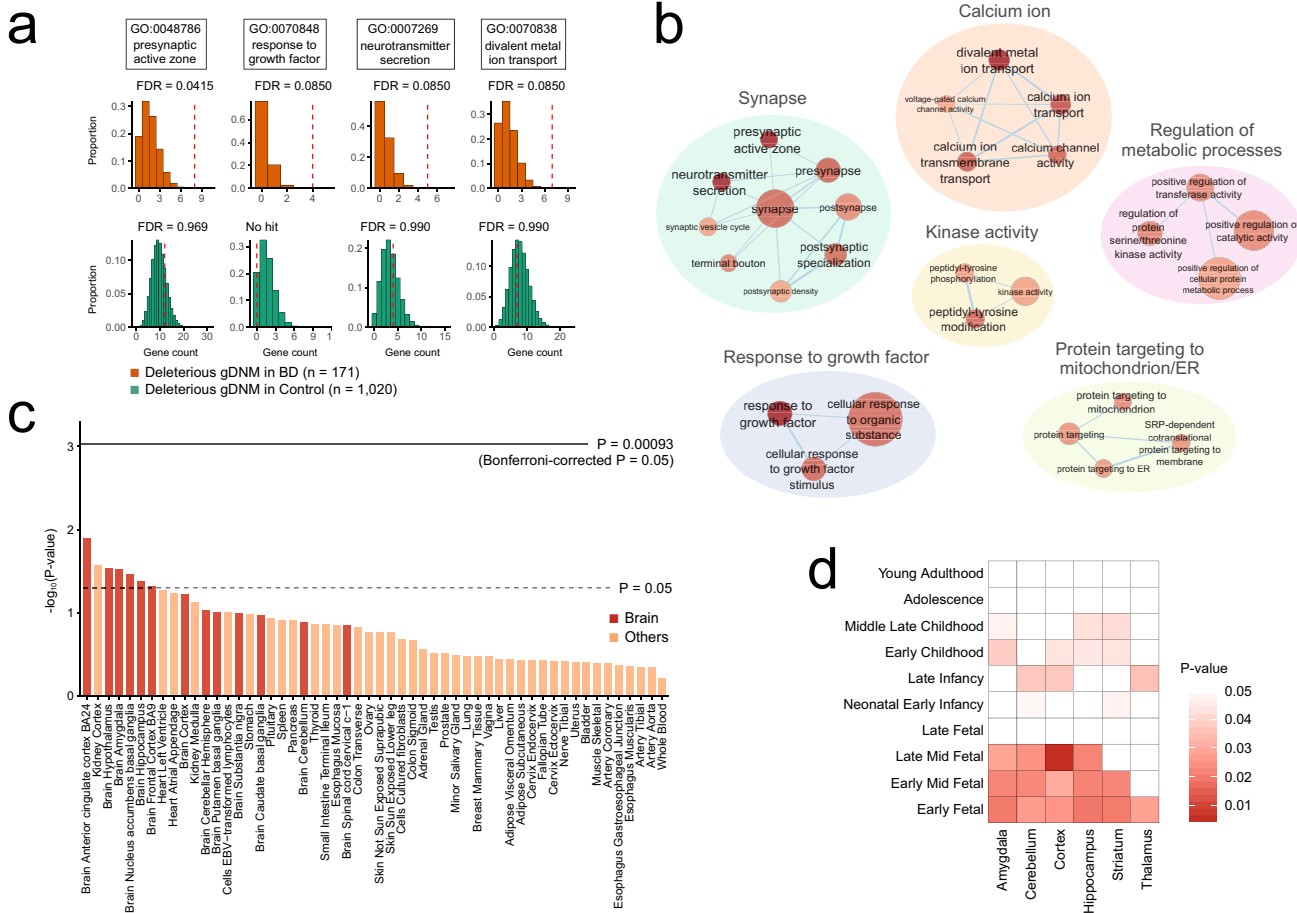

**Fig. 2 Gene set enrichment analysis of deleterious gDNMs in BD. a** Gene ontology (GO) terms enriched among the genes hit by deleterious (LoF or damaging missense/inframe indel) gDNMs in BD at a false discovery rate (FDR) <0.1. FDR adjustment was performed based on the number of terms in the corresponding GO category (biological process, cellular component, or molecular function). Uncorrected P values for our observations (the red dotted lines) were calculated by DNENRCIH that considers confounding factors, such as gene sizes and local sequence contexts ("Methods"). The histograms indicate the distributions of the expected number of gDNMs hitting the corresponding GO term (the x axis), which was generated by one million random permutations by DNENRCIH. Top, BD; bottom, controls. **b** Network visualization of the GO terms enriched among the genes hit by deleterious gDNMs in BD at uncorrected $P < 0.05$. The node colors and label sizes indicate the statistical significance of enrichment (deep red indicates the most significant terms). The node sizes indicate the number of genes included in a term. The edge width is proportional to the overlap coefficient. The full list of GO terms with uncorrected $P < 0.05$ is shown in Supplementary Data 2. **c** Enrichment analysis of the genes hit by deleterious gDNMs in BD for the top 2% of the genes with the highest expression in each of the 54 human tissues in the GTEx dataset. Uncorrected P values calculated by DNENRICH are shown as bar plots. The dotted and solid lines indicate the nominal ($P = 0.05$) and the Bonferroni-corrected ($P = 0.00093 = 0.05/54$) significance threshold, respectively. Bars are color coded as shown on the lower right of the plot. **d** Enrichment analysis of the genes hit by deleterious gDNMs in BD for the top 2% of the genes with the highest expression in each developmental period of a brain region in the BrainSpan Human Developmental Transcriptome dataset. Cells are color coded by uncorrected P values calculated by DNENRICH as shown on the right of the grid cells.

DNENRICH analysis considering gene sizes and other confounding factors (Fig. 3e, $P = 0.00467$), whereas there was no significant enrichment among the genes hit by deleterious gDNMs in controls from ref. [20] ($P = 0.334$). Genes most characteristic of the cluster 8 cells include *CHD5*, a gene encoding a chromodomain helicase protein, and *SNORD115*, encoding small nuclear non-coding RNAs in the locus imprinted in Prader–Willi syndrome[36] (Fig. 3f). GO enrichment analysis of the c8 signature genes showed overrepresentation of synaptic and ion channel genes (Fig. 3g and Supplementary Data 4). Enrichment of these GO terms remained significant even after excluding the genes hit by deleterious gDNMs in BD from the input list of the c8 signature genes (Supplementary Data 4). Therefore, enrichment of these GO terms is not solely explained by direct targets of deleterious gDNMs in BD, which also showed enrichment of the same or similar terms in the GO enrichment analysis in Fig. 2a, b. Another interesting finding is that the c8 signature genes

are significantly overlapped with known DD genes[37] ($P = 7.09 \times 10^{-7}$, hypergeometric test, Fig. 3h and Supplementary Data 5). We also noticed that the c8 signature genes include two genes (*CACNA1C* and *ANK3*) significantly associated with BD in a large-scale GWAS[4]. Again, this is an observation unlikely to have occurred by chance ($P = 0.0210$, hypergeometric test, Fig. 3h).

**Genes recurrently hit by deleterious gDNMs.** Previous studies of gDNMs in other neuropsychiatric disorders have discovered a number of robust disease-associated genes by identifying those recurrently hit by deleterious gDNMs[13–20]. Among the genes with deleterious gDNMs in BD, we found that two genes, *XKR6* and *MRC2*, are recurrently hit by deleterious gDNMs. The extreme mutational constraint in these genes (pLI > 0.99) further supports their association with a phenotype characterized by reduced fecundity, such as BD and other neuropsychiatric

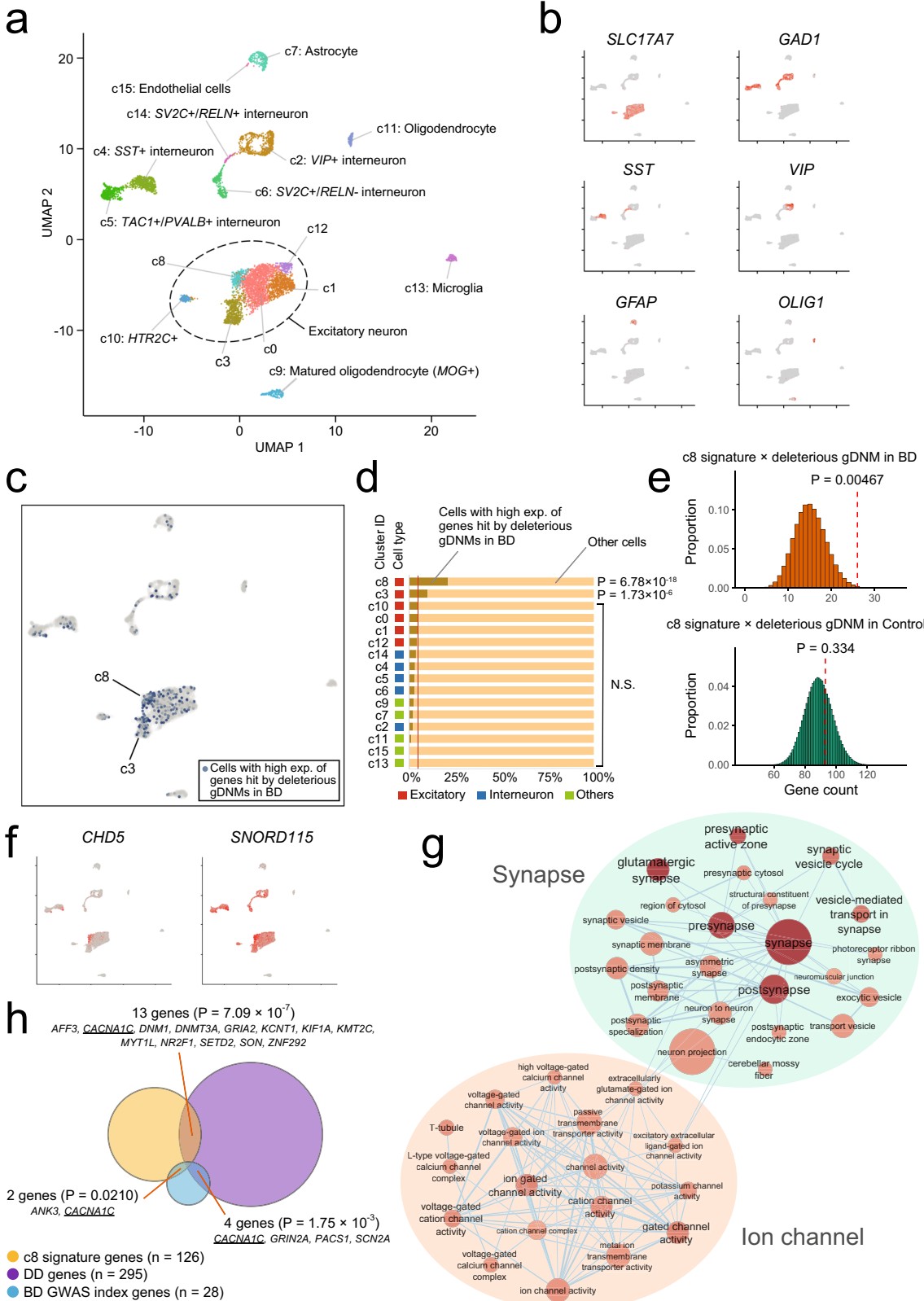

disorders[38]. However, when we performed a statistical assessment based on an established model of per-gene gDNM rates[39], the observed numbers of deleterious gDNMs in these genes did not reach the exome-wide significance in BD (defined as $P = 0.05/18,271 = 2.74 \times 10^{-6}$, based on the number of protein-coding genes with an available mutation rate in ref. [39]; Table 1). Though

we failed to identify a single gene whose deleterious gDNMs are specifically associated with BD, we next investigated whether deleterious gDNMs in *XKR6* or *MRC2* are reported in large-scale studies of other neuropsychiatric/developmental disorders[13,15,16,20,37] (Supplementary Table 1). We found that there are multiple deleterious gDNMs in *XKR6* and *MRC2* in

**Fig. 3 Single-cell (nucleus) enrichment analysis of the genes hit by deleterious gDNMs in BD. a** UMAP representation of 16 cell clusters (c0–15) identified from single-nucleus RNA sequencing data of human adult anterior cingulate cortices. Cell clusters are annotated based on the patterns of marker gene expression (**b** and Supplementary Fig. 3). **b** Expression patterns of representative marker genes for excitatory neurons (*SLC17A7*), general and specific interneurons (*GAD1*, *SST*, and *VIP*), astrocytes (*GFAP*), and oligodendrocytes (*OLIG1*). **c** A plot of the top 5% of the cells preferentially expressing genes hit by deleterious gDNMs in BD onto the UMAP. The blue and gray dots indicate the top 5% cells and the others, respectively. **d** Proportions of cells preferentially expressing the genes hit by deleterious gDNMs in BD in each cluster. Cluster IDs and cell types (red: excitatory, blue: interneuron, and light green: other cells) are shown on the left of the bar plots. The red vertical line indicates the theoretical expectation (5%). *P* values are calculated by comparing the observed and expected proportions (one-tailed binomial test with Bonferroni correction) are shown on the right of the bar plots. N.S. not significant. **e** Enrichment analysis of the genes hit by deleterious gDNMs in BD (top, orange) or controls (bottom, green) for the c8 signature genes. Uncorrected *P* values were calculated by DNENRICH that considers confounding factors, such as gene sizes and local sequence contexts ("Methods"). **f** Plots of the expression patterns of representative genes upregulated in the cluster 8 (c8 signature genes) onto the UMAP. **g** Network visualization of the GO terms significantly enriched among the c8 signature genes (FDR < 0.05). The node colors and label sizes indicate the statistical significance of enrichment (deep red indicates the most significant terms). The node sizes indicate the number of genes included in a term. The edge width is proportional to the overlap coefficient. Nodes are connected when the overlap coefficient of the containing genes >0.5. **h** A Venn diagram showing overlaps between the c8 signature genes and known DD genes[37] or the index genes in the largest BD GWAS to date[4]. Uncorrected *P* values for the observed overlaps under the hypergeometric distribution (one tailed) and the symbols of the overlapping genes are shown. *CACNA1C* is underlined as the only one gene in the intersection of three gene sets: c8 signature, DD, and BD GWAS index genes.

other disorders, whereas no such gDNMs were observed in controls (Table 1). In particular, we observed that deleterious gDNMs in *XKR6* (2 LoF and 9 damaging missense gDNMs) are exome-wide significantly enriched in the combined group of BD, schizophrenia, ASD, and DD (Table 1, $P = 2.24 \times 10^{-6}$, one-tailed Poisson test), indicating a strong association of *XKR6* with a broad spectrum of neuropsychiatric/developmental disorders.

Overall enrichment of LoF gDNMs in constrained genes indicates their contribution to BD pathogenesis (Fig. 1c, d). While there are no constrained genes recurrently hit by LoF gDNMs in BD probands included in this study, we found that there are eight constrained genes hit by LoF gDNMs in a broad spectrum of neuropsychiatric/developmental disorders, including BD (Table 1). Among them, we observed exome-wide significant enrichment of LoF gDNMs in *KMT2C*, a gene encoding a histone methyltransferase protein, in the combined group of neuropsychiatric/developmental disorders ($P = 2.48 \times 10^{-7}$ for BD + schizophrenia + ASD, and $P = 3.03 \times 10^{-16}$ for BD + schizophrenia + ASD + DD, one-tailed Poisson test). We also found enrichment of LoF gDNMs in *SMARCC2*, encoding a component of the SWI/SNF chromatin remodeling complexes, with significance close to the exome-wide threshold in the combined group of BD, schizophrenia, and ASD ($P = 5.62 \times 10^{-6}$, one-tailed Poisson test). Therefore, LoF gDNMs in these genes can be, in a general sense, considered as those associated with a broad spectrum of neuropsychiatric/developmental disorders at a certain level of significance.

**Systematic analysis of postzygotic DNMs (pzDNMs) in BD.** Our analysis of genes recurrently hit by deleterious gDNMs identified *KMT2C* as a gene whose LoF gDNMs are robustly associated with a broad spectrum of neuropsychiatric/developmental disorders. This gene was reported as a gene causal for neurodevelopmental disorders including Kleefstra syndrome phenotypic spectrum, which is characterized by ASD, intellectual disability (ID), facial dysmorphisms, and childhood hypotonia[40–42]. On the other hand, the BD case with an LoF gDNM of *KMT2C* (p.Lys3601*) had no history of developmental delay but rather received a higher education, indicating that there is a clear phenotypic discrepancy. Plotting of the gDNMs in our and previously reported cases shows that there are multiple LoF gDNMs observed in DD/ASD cases downstream of the p.Lys3601* variant observed in BD (Fig. 4a). The p.Lys3601* variant is located at an essential (i.e., non-isoform-specific) exon of *KMT2C*. Therefore, this nonsense variant in BD is highly likely to cause loss of gene function. Based on this phenotypic discrepancy,

we re-examined the mapped reads and the Sanger sequencing chromatogram supporting the p.Lys3601* variant. We found that the intensity of the variant allele is apparently lower than that of the reference allele (Fig. 4b, left). Subsequent direct sequencing of the PCR fragments containing the p.Lys3601* variant and a nearby common heterozygous SNP (rs74483926: g.151859683G>A) confirmed that the p.Lys3601* variant is, in fact, a pzDNM (Fig. 4b, middle). We verified that the p.Lys3601* pzDNM is not a variant due to clonal hematopoiesis of indeterminate potential (CHIP), by identifying the same pzDNM in the DNA samples derived from saliva, nail, and hair (Fig. 4b, right).

Being inspired by this observation, we systematically investigated pzDNMs in BD, primarily hypothesizing that pzDNMs of DD genes may contribute to neuropsychiatric disorders such as BD by affecting specific subsets of brain cells. We included our own BD trios in this analysis considering the availability of the original DNA samples for validation experiments. By performing a re-analysis of the exome data with Mutect2[43] followed by target amplicon sequencing (TAS) of all the detected candidates[44] ("Methods"), we identified a total of 28 validated pzDNMs not observed in the general population (Supplementary Data 6 and Supplementary Fig. 4a). We also found that three variants called as a gDNM, including the *KMT2C* p.Lys3601* variant, are highly likely to be a pzDNM (Supplementary Data 1 and 6), while such pseudo-gDNM calls represent the minority of all gDNM calls (Supplementary Fig. 4b) and should have no major effect on our analysis of the overall profile of gDNMs in BD (in Figs. 1–3).

The validated 28 pzDNMs includes 16 deleterious (LoF or damaging missense) variants. Of these, four deleterious pzDNMs hit known DD genes[37]. We then statistically assessed this observation. After confirming similar patterns of base substitutions across gDNMs and pzDNMs (Supplementary Fig. 4c), we calculated the expected proportion of deleterious pzDNMs hitting the 299 known DD genes to all deleterious pzDNMs as 0.0317 (the dotted lines in Fig. 4c) by summing the per-gene rates for deleterious gDNMs[39] ("Methods"). When compared with this theoretical expectation, the observed proportion (0.25; 4 out of the 16 deleterious pzDNMs hit the DD genes) is highly unlikely to be a chance finding (Fig. 4c, left, $P = 0.00135$, one-tailed binomial test), supporting our hypothesis that deleterious pzDNMs of DD genes are associated with BD. On the other hand, there was no enrichment of known DD genes among the genes hit by deleterious gDNMs in BD (Fig. 4c, right, $P = 0.885$, *n* of deleterious gDNMs = 160 [not including the 8 inframe gDNMs and the 3 confirmed pzDNMs], see "Methods" for details). By

**Table 1 Genes with recurrent deleterious gDNMs in BD and other neuropsychiatric/developmental disorders.**

| Gene | pLI | gDNM count[a] | | | | | gDNM enrichment | | | | |
| | | BD (N = 354) | SCZ (N = 2839) | ASD (N = 6430) | DD (N = 31,058) | Control (N = 2179) | BD (N = 354) Uncorrected P value[b] | BD + SCZ + ASD (N = 9623) gDNM count total[a] | Uncorrected P value[b] | BD + SCZ + ASD + DD (N = 40,681) gDNM count total[a] | Uncorrected P value[b] |
|---|---|---|---|---|---|---|---|---|---|---|---|
| *Genes with multiple deleterious gDNMs in BD* | | | | | | | | | | | |
| XKR6 | 0.9971 | 2 (0.2) | 1 (0.1) | 1 (1.0) | 7 (1.6) | 0 | $1.13 \times 10^{-4}$ | 4 (1.3) | $8.58 \times 10^{-4}$ | 11 (2.9) | $\mathbf{2.24 \times 10^{-6}}$ |
| MRC2 | 0.9964 | 2 (0.2) | 0 | 0 | 3 (3.0) | 0 | $5.47 \times 10^{-4}$ | 2 (0.2) | 0.231 | 5 (3.2) | 0.341 |
| *Constrained genes with LoF gDNMs in BD and other disorders* | | | | | | | | | | | |
| KMT2C | 1 | 1 | 2 | 4 | 14 | 0 | 0.0148 | 7 | $\mathbf{2.48 \times 10^{-7}}$ | 21 | $\mathbf{3.03 \times 10^{-16}}$ |
| SMARCC2 | 1 | 1 | 0 | 2 | 0 | 0 | 0.00404 | 4 | $5.62 \times 10^{-6}$ | 4 | 0.00135 |
| XPO4 | 1 | 1 | 0 | 1 | 1 | 0 | 0.00378 | 2 | 0.00496 | 3 | 0.00997 |
| TNRC18 | 0.9999 | 1 | 0 | 1 | 0 | 0 | 0.00542 | 2 | 0.00991 | 2 | 0.130 |
| KLF4 | 0.9743 | 1 | 0 | 0 | 1 | 0 | 0.00125 | 1 | 0.0334 | 2 | 0.00937 |
| ATP2B2 | 0.9997 | 1 | 0 | 0 | 1 | 0 | 0.00330 | 1 | 0.0861 | 2 | 0.0564 |
| CCAR1 | 0.9999 | 1 | 0 | 0 | 1 | 0 | 0.00436 | 1 | 0.112 | 2 | 0.0909 |
| RAPGEF2 | | 1 | 0 | 0 | 1 | 1 | 0.00458 | 1 | 0.117 | 2 | 0.0988 |

Boldface indicates genes with exome-wide significance defined as $P < 2.74 \times 10^{-6}$ ($= 0.05/18,271$).

ASD autism spectrum disorder, BD bipolar disorder, DD developmental disorder, gDNM germline de novo mutation, LoF loss of function, pLI probability of being LoF-intolerant, SCZ schizophrenia.

[a]For genes with multiple deleterious gDNMs in BD and other disorders, the total numbers of deleterious gDNMs with the numbers of LoF and damaging missense/inframe indel gDNMs in the parenthesis are shown. For constrained genes with LoF gDNMs in BD and other disorders, the numbers of LoF gDNMs are shown.

[b]For genes with multiple deleterious gDNMs in BD and other disorders, P values were calculated by comparing the observed and expected numbers of deleterious gDNMs. For constrained genes with LoF gDNMs in BD and other disorders, P values were calculated by comparing the observed and expected numbers of LoF gDNMs. The listed genes are limited to those also hit by deleterious/LoF gDNMs in other neuropsychiatric/developmental disorders than BD (i.e., SCZ, ASD, or DD in this Table, Supplementary Table 1).

assuming that disease-associated pzDNMs would be observed in the general population in a peripheral tissue-specific manner, we also performed an analysis not excluding pzDNMs same to the variants observed in the general population (gnomAD and ToMMo). Again, we confirmed that the proportion of deleterious pzDNMs in BD hitting known DD genes is higher than the expectation (Supplementary Fig. 4d, left, $P = 0.00120$, one-tailed binomial test), whereas there was no such enrichment in deleterious gDNMs in BD (Supplementary Fig. 4d, right, $P = 0.863$).

Regarding the individual genes hit by deleterious pzDNMs in BD, it is noteworthy that two BD probands carry a deleterious pzDNM in the same gene, *SRCAP* (Fig. 4d), while one of them (p. Arg971Cys) is same to a variant observed in gnomAD. The *SRCAP* gene encodes a SNF2-related chromatin-remodeling ATPase[45,46] and known as the causative gene for Floating–Harbor syndrome[47]. An exome-wide simulation analysis randomly generating 26 deleterious pzDNMs (i.e., the observed number of deleterious pzDNMs including those observed in the general population in our BD cohort) demonstrated that it is unlikely to observe two probands carrying a deleterious pzDNM in the same gene (exome-wide simulation $P = 0.0344$). Both of the deleterious pzDNMs (p.Leu696Phe and p.Arg971Cys) are missense variants under strong evolutionary constraint (Fig. 4d) and having a Combined Annotation Dependent Depletion (CADD) score >25 (i.e., top ~0.3% of the variants predicted to be damaging by CADD). Specifically, p. Leu696Phe is predicted to be damaging by all seven algorithms assessing the functional impact of missense variants (see "Methods" and Supplementary Data 6). We subsequently assessed the mutational effect of these pzDNMs on the SRCAP protein. The SRCAP protein is a subunit of a huge multiprotein complex, the SRCAP complex, which deposits the histone variant H2A.Z into chromatin in an ATP-dependent manner. The cryo-electron microscopic structures of the human SRCAP complex, as well as that of the yeast homolog (the SWR1 complex) bound with nucleosome, have been determined[48,49]. Leu696 and Arg971 of the human SRCAP protein are structurally conserved in the yeast SWR1 protein and correspond to Leu774 and Lys1069 of the yeast SWR1 protein, respectively, Fig. 4e, f). Leu696, located in lobe 1 of the ATPase motor domain of human SRCAP, is disordered in the structure (Fig. 4e). On the other hand, in the yeast SWR1 complex structure, lobe 1 directly interacts with the bound nucleosome, making an extensive interface with the two gyres of DNA around superhelical location +2 (SHL +2) and SHL −6 (Fig. 4f, g). Leu774 in the yeast SWR1 complex forms van der Waals interactions with nearby residues such as Leu786, Trp790, Ala795, and Phe796, stabilizing the local conformation of the loop–helix–loop element containing Arg787 and Asn791, which in turn interacts with nucleosomal DNA at SHL −6 (Fig. 4g). Structural modeling using the yeast SWR1 protein shows that substituting Leu774 with Phe results in a slight steric clash with Ala795 and Phe796, suggesting this substitution (and possibly p.Leu696Phe in human SRCAP protein) may affect the stability of the protein. Besides, a small conformational change caused by this substitution may diminish the histone deposition activity of the SRCAP complex by affecting the interaction between lobe 1 and the two gyres of nucleosomal DNA. Both human Arg971 and yeast Lys1069 are located in the antiparallel α-helices that connect the Arp6 subunit with the rest of the complex (Fig. 4e, f). Their side chains make no inter- or intra-molecular interactions and are protruding to the solvent. This observation suggests that the p.Arg971Cys substitution may not compromise the protein stability or nucleosome-binding activity by itself. However, the antiparallel α-helices have been proposed to have a regulatory role yet to be discovered[48], and the

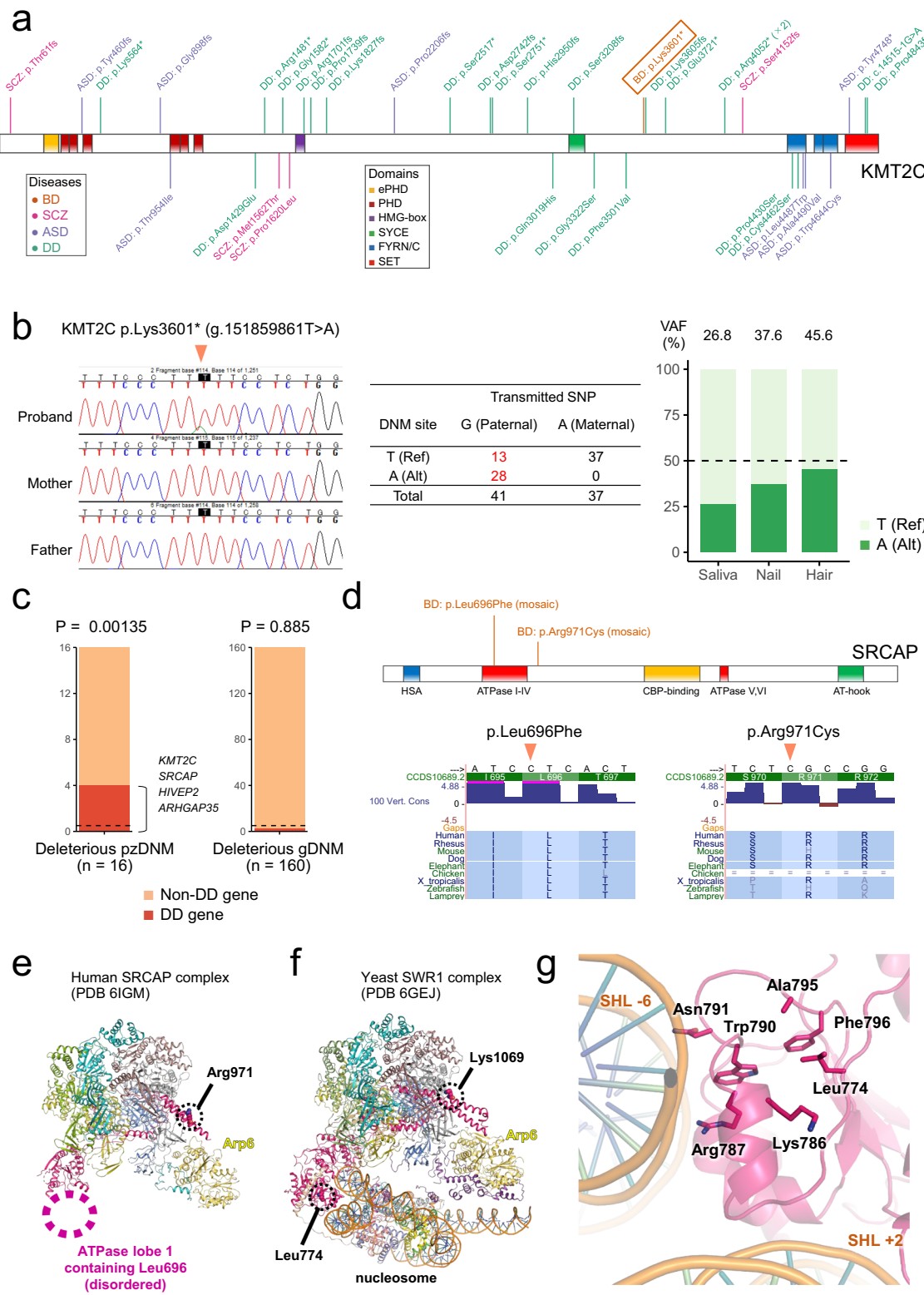

p.Arg971Cys substitution may indirectly affect the cellular function of SRCAP by impeding such a process.

Taken these data together, *SRCAP* can be considered as a prominent candidate gene whose deleterious pzDNMs are associated with BD, while further investigation is required. We also note that our extensive literature search and curation of pzDNMs in non-psychiatric individuals demonstrate that *SRCAP*

cannot be a gene frequently mutated in clonally expanded hematopoietic cells (Supplementary Note 1).

## Discussion

We in this study systematically investigated both gDNMs and pzDNMs in coding regions in BD. To our knowledge, this is the largest study of DNMs in BD.

**Fig. 4 Systematic analysis of pzDNMs in BD and deleterious pzDNMs in *KMT2C* and *SRCAP*. a** Schematic representation of the domain structure of the KMT2C protein and the positions of DNMs reported in our and previous studies (see Supplementary Table 1). LoF and missense DNMs are shown on the upper and lower sides, respectively. The DNMs are color coded according to the diagnostic group. Kleefstra syndrome phenotypic spectrum is included in DD. *SCZ* schizophrenia. **b** A confirmed LoF (p.Lys3601*) pzDNM in *KMT2C* in BD. Left: Sanger sequencing chromatogram confirming the de novo p. Lys3601* variant in the BD proband. The intensity of the variant allele is lower than that of the reference allele. Middle: the allele distribution of the p. Lys3601* (g.151859861T>A) variant and a nearby heterozygous common SNP (rs74483926, g.151859683G>A) in the cloned PCR fragments from the BD proband. Red letters indicate mosaicism of the p.Lys3601* variant in the paternal allele. Right: variant allele fractions of the p.Lys3601* variant in the saliva, nail, and hair of the BD proband shown as a bar plot. This observation excludes the possibility that the p.Lys3601* pzDNM is due to clonal hematopoiesis of indeterminate potential (CHIP). **c** Proportions of the deleterious (LoF or damaging missense) pzDNMs (left) and gDNMs (right) not observed in the general population in BD hitting a known DD gene[37]. The dotted lines indicate the theoretical expectation based on an established mutational model[39]. *P* values calculated by a comparison between the observation and the expectation (one-tailed binomial test) are shown above the bars. **d** The two pzDNMs (p.Leu696Phe or p.Arg971Cys) in *SRCAP* observed in two different BD probands. Top: schematic representation of the domain structure of the SRCAP protein[47] and the positions of the observed pzDNMs. Bottom: the evolutionary conservation around the pzDNM sites from the Multiz Alignments of 100 Vertebrates. For the amino acid hit by the p.Arg971Cys variant, no missense variant introducing a substitution to cysteine was observed. **e** Structure of human SRCAP complex. The location of the ATPase lobe 1 containing Leu696 had weaker density and was thus omitted from the deposited coordinates. **f** Structure of yeast SWR1 complex bound with nucleosome. Leu696 in human SRCAP corresponds to Leu774 in yeast SWR1. Arg971 in human SRCAP is conservatively substituted with Lys (Lys1069) in yeast SWR1. **g** A close-up view of Leu774 and nearby residues in the yeast SWR1 complex. The positions of the two DNA gyres (SHL +2 and SHL −6) are shown. The residues involved in a hydrophobic core with Leu774 (corresponding to Leu696 in human SRCAP complex) and a recognition of the SHL −6 position of DNA are shown as sticks with nitrogen atoms in blue.

Our analysis of patterns of enrichment of gDNMs in BD demonstrates that gDNMs predicted to be deleterious to the gene function (i.e., LoF or damaging missense/inframe indel), not in the general population, and hitting a constrained gene are more frequent in BD, especially in BD subtypes that are generally considered to be severe (i.e., BDI or SCZAD; Fig. 1). These patterns are similar to the observation in studies of gDNMs in other neuropsychiatric disorders[15,17,29,39]. Therefore, like other neuropsychiatric disorders, deleterious gDNMs should explain a part of the genetic structure of BD, and the interpretable results in our analysis support the overall validity of this study.

Gene set enrichment analysis indicates that the genes hit by deleterious gDNMs in BD are enriched for genes localized to synapse, involved in calcium ion transport, and regulating the response to growth factors (Fig. 2a, b). Among them, synaptic and calcium ion channel genes have been consistently implicated in BD by multiple lines of evidence from genetic, pharmacological, electrophysiological, and biochemical studies of BD[3,4,50–53]. Also, there are studies suggesting the involvement of the growth factor pathway in the pathophysiology of BD and other neuropsychiatric disorders[54,55]. Specific genes in this pathway and hit by a deleterious gDNM in BD include *PDGFB*, a platelet-derived growth factor gene responsible for idiopathic basal ganglia calcification[56] that frequently accompanies psychiatric symptoms such as mood instability[57], and *PLCG1*, encoding phospholipase C gamma 1 whose forebrain-specific disruption causes mania-like behavior in mice[58]. Our results further consolidate the involvement of synaptic and calcium ion channel genes and provide additional support that genes related to response to growth factors may play a role in the etiology of BD.

An analysis of the genes hit by deleterious gDNMs in BD integrating transcriptome data of bulk tissues shows an expected enrichment of genes highly expressed in the brain, with a prenatal bias (Fig. 2c, d). Subsequent analysis being informed by single-cell RNA sequencing data of ACC identifies a subset of excitatory neurons preferentially expressing the genes hit by deleterious gDNMs in BD (Fig. 3c–e). While this would be too hypothetical at this moment, disruption of these neurons that are also characterized by high expression of synaptic, ion channel, and DD genes (Fig. 3g, h) may play a critical role in the BD pathophysiology. Also, our analysis represents an example of how we can integrate results of rare variant studies with single-cell RNA sequencing data.

Besides the elucidation of biological pathways and brain regions/cells involved in the disease etiology, another major aim of genetic studies is to identify specific genes robustly associated with a disease. To this end, we performed a statistical analysis of genes recurrently hit by deleterious DNMs. While we failed to identify genes reaching the exome-wide significance threshold in the BD-only analysis, we observed enrichment of deleterious gDNMs in *XKR6*, *KMT2C*, and *SMARCC2* in a broad spectrum of neuropsychiatric/developmental disorders, including BD, at a certain level of statistical significance (Table 1). *XKR6* encodes a member of X Kell blood group precursor-related family proteins whose function is largely uncharacterized. There is no human disease associated with this gene in the Online Mendelian Inheritance in Man database, indicating that *XKR6* is a novel neuropsychiatric/developmental disorder gene. GWAS of common SNPs identified genome-wide significant associations between the locus including *XKR6* and two neuropsychiatric traits: neuroticism[59] and risk tolerance[60]. These associations suggest this gene's role in mood regulation and risk-taking behavior, both of which are BD-related phenotypes. The other genes, *KMT2C* and *SMARCC2*, are both involved in chromatin modification and are known as genes causative for severe neurodevelopmental disorders (*KMT2C* for Kleefstra syndrome phenotypic spectrum[40,41] and *SMARCC2* for Coffin–Siris syndrome[61]). Consistent with our results indicating association of *KMT2C* and *SMARCC2* with a broad spectrum of neuropsychiatric/developmental disorders, the original studies discovering these genes as disease causative reported that there is considerable phenotypic variability, such as mild-to-severe ID, among the mutation carriers[40,41,61]. This large phenotypic variability can be explained by factors, including genotype–phenotype relationships, genetic backgrounds, epistases, and environmental influences; however, in the case of BD proband with an LoF variant in *KMT2C*, the observed milder phenotype can be most reasonably explained by the fact that this variant is a pzDNM (Fig. 4b). Supporting this, in the study discovering a causal relationship between *KMT2C* and Kleefstra syndrome phenotypic spectrum, it was reported that an individual with the mildest phenotypes in their study carries a deleterious pzDNM of this gene[41]. In addition, a study screened for neuropsychiatric phenotypes in parents of Kleefstra syndrome phenotypic spectrum probands reported that the parents with a postzygotic mosaic deletion of *EHMT1*, another chromatin-modifying gene responsible for Kleefstra syndrome phenotypic spectrum, fulfilled the criteria for ASD and major depressive disorder[62]. One of the parents with a postzygotic mosaic deletion of *EHMT1* has psychotic symptoms along with depressive episodes. These individual examples further support a hypothesis that deleterious pzDNMs of

DD genes would contribute to milder neuropsychiatric phenotypes by affecting specific subsets of brain cells. To statistically test this notion, we performed a systematic analysis of pzDNMs in BD and observed that deleterious pzDNMs in DD genes are significantly enriched when compared with the theoretical expectation (Fig. 4c). Together with the enrichment of genes hit by deleterious gDNMs in a subset of excitatory neurons preferentially expressing DD genes (Fig. 3h), our results indicate that dysfunction of DD genes in a specific population of brain cells would contribute to the pathogenesis of BD.

Among the genes hit by pzDNMs in BD, it is notable that two deleterious pzDNMs in BD hit the same *SRCAP* gene. This observation is unlikely to have occurred by chance (P from exome-wide simulation = 0.0344). There should be two possible explanations for this result: deleterious pzDNMs in *SRCAP* are indeed contributing to the risk of BD, or *SRCAP* is a gene frequently mutated in clonally expanded hematopoietic cells. To evaluate the latter possibility, we performed an extensive literature search and curation of pzDNMs in non-psychiatric individuals (Supplementary Note 1). We conclude that this is unlikely the case, while we cannot completely exclude this possibility as we could not access the DNA samples from non-hematopoietic cells of the BD probands with a deleterious pzDNM in *SRCAP*. The *SRCAP* gene is highly intolerant to LoF variants in the general population (pLI > 0.99) and known as a gene responsible for Floating–Harbor syndrome[47], which is characterized by ID, short stature, expressive-language delay, and distinctive facial appearance. In Floating–Harbor syndrome, pathogenic gDNMs are strongly clustered in the last exon, suggesting a dominant-negative disease mechanism[37,47]. On the other hand, multiple nonsense-mediated mRNA decay-inducible LoF gDNMs (i.e., those not in the last exon or the last 50 bases of the penultimate exon) were observed in ASD or ID cases[17,63–65]. Therefore, heterozygous dominant-negative and LoF variants would be linked to Floating–Harbor syndrome and ASD/ID, respectively. The two observed deleterious pzDNMs in *SRCAP* in our BD samples, the p.Leu696Phe and p.Arg971Cys variants, are both at an amino acid highly conserved across species (Fig. 4d). Structural consideration using the yeast SWR1 complex as a homologous model showed that the p.Leu696Phe variant hit lobe 1 of the ATPase motor domain, probably affecting the stability or the histone deposition activity of the human SRCAP. Thus, the p.Leu696Phe variant, not observed in the general population, is predicted to cause a deleterious effect similar to LoF. On the other hand, p.Arg971Cys variant cannot be confidently evaluated by the structure modeling and exist in the non-neuro population of gnomAD at a low frequency (minor allele frequency = $9.61 \times 10^{-6}$). Therefore, while the statistical significance of observing two deleterious pzDNMs in *SRCAP* and the predicted deleteriousness of the p.Leu696Phe variant indicates this gene as a good candidate gene whose deleterious pzDNMs are associated with BD, there is some ambiguity especially about the pathogenicity of the p.Arg971Cys variant.

Besides this potential uncertainty for pzDNMs in *SRCAP*, we are aware that the present study has several other limitations. First, while this is the largest study of DNMs in BD to date, the sample size is still modest when compared with the studies for other disorders such as ASD and schizophrenia[13,15–17,20]. Second, we in this study included the cases of SCZAD to prioritize the increase of sample size and statistical power. While studies have supported genetic, phenotypic, and physiological similarities between BD and SCZAD[21–26] and thereby previous genetic investigations of BD often include SCZAD individuals[9,24,66], there might be controversy whether it is justifiable to include SCZAD in a study of BD. Other potential limitations include that we combined the data from different ethnicities in this study. However, it is indicated that gDNM rates are consistent across

different ethnicities[15,19]. Indeed, previous studies of gDNMs comparing different ethnicities of cases and controls have reported similar gDNM rates for synonymous gDNMs across ethnicities[15,19]. Thus, our results are unlikely to be critically affected by this potential limitation. Another thing is that we considered unaffected siblings of ASD children as controls. These individuals may not be ideal controls given the common age of onset for BD. However, we expect that the impact of this potential limitation is minimal considering the low prevalence of BD[67]. Also, this cannot cause the overestimation of enrichment of deleterious gDNMs in BD. Therefore, we can conclude that the major limitations of this study are related to the sample size. Nevertheless, we would like to emphasize that several findings in our study remain significant after correction for multiple testing.

In summary, we conducted the largest study of DNMs in BD and demonstrate that specific types of gDNMs and pzDNMs contribute to the genetic architecture of BD. Also, our analysis identifies genes recurrently hit by deleterious gDNMs or pzDNMs and provides a variety of insights into the neurobiology of BD. Specifically, our identification of a subset of excitatory neurons characterized by high expression of the genes hit by deleterious gDNMs in BD as well as synaptic, calcium ion channel, and known DD genes and demonstration of the enrichment of deleterious pzDNMs in DD genes in BD may both allow pinpointing of the brain regions/cell types playing a pivotal role in BD. Investigation with additional samples including postmortem brain tissues derived from donors with BD will further advance our understanding of the contribution of pzDNMs. In addition to scaling up the sample size to conquer the major limitations of this study, by functionally characterizing the identified candidate disease-associated neuronal cells and detecting brain regions/cell types affected by deleterious pzDNMs of DD genes in BD postmortem brains, we would be able to approach the discovery of the neural circuit(s) responsible for mood stabilization[68] and the elucidation of the fundamental neuropathology of BD.

## Methods

**Study participants**. We recruited probands with BD and their parents (trios) through Bipolar Disorder Research Network Japan (http://bipolar.umin.jp/) or the participating institutions. All the probands had been diagnosed as BD or SCZAD by trained psychiatrists. Their diagnoses were further verified based on Diagnostic and Statistical Manual of Mental Disorders -IV, -IV-TR, or -5, using M.I.N.I. (Mini International Neuropsychiatric Interview)[69] by a psychiatrist or psychologist or using semi-structured interviews by a senior psychiatrist (T.K.). All the parents were screened for mental disorders using M.I.N.I. except for some cases for whom interviews were done using printed matter because of disabilities. Additional questions to verify the past history of major depressive episodes were asked to the participants. Detailed procedures are described in our previous study[7]. Participants with BDI (N = 115) were prioritized for sequencing. We included ten families with parents diagnosed as BD, SCZAD, or schizophrenia to increase the number of trios. After a detailed explanation and obtainment of written informed consent, blood or saliva of the participants was collected. A psychologist or a psychiatrist verified that the participants are capable of informed consent through the interviews. A total of 171 probands were recruited with their parents, in which 97 probands (56.7%) were female. The average age of recruitment is 36.0 ± 9.5 years (standard deviation) and the average age of disease onset is 22.9 ± 7.5 years. The DNA samples from blood were extracted by standard procedures. For the saliva samples, we used the Oragene® DISCOVERY kit (DNA Genotek, Ottawa, ON, Canada). All the samples reported in our previous study[7] were included in this study. This study was designed according to the Helsinki declaration and approved by RIKEN Wako Research Ethics First Committee, The Ethical Review Board of Juntendo University Faculty of Medicine, Yokohama City University Human Genome and Gene Research Ethics Committee, Ethical Committee of Saitama Medical University, and the Ethical Review Boards for Human Genome Studies at Fujita Health University. No statistical methods were used to predetermine sample sizes. We recruited BD trios as much as possible. Randomization is not applicable, because this study is not a clinical intervention study.

**Whole-exome sequencing (WES)**. Exome enrichment was performed with the Agilent SureSelect Human All Exon v4/5/6 kits (Agilent Technologies, Santa Clara, CA, USA) according to the manufacturer's instruction. The prepared DNA libraries underwent sequencing by HiSeq2000/2500 (Illumina, San Diego, CA,

USA) with paired-end 101 bp reads. One sample was re-sequenced by Nova-Seq6000 (Illumina) due to the low quality in the initial sequencing. Library preparation and sequencing were performed blindly to the affected status of the participants. No replication was done for the same sample.

**External data**. The exome sequencing data of 86 SCZAD trios[13] were obtained from dbGaP after the authorization process (accession number phs000687.v1.p1; project number 19482). The exome sequencing data of families with ASD in Simons Simplex Collection (SSC)[18,70] were obtained from SFARI base after receiving an approval (accession number SSC WES3; SFARI ID: 2382.1.2). The list of exonic gDNMs identified in a whole-genome sequencing study of 97 BD trios was obtained from Supplementary Table 1 of the Goes et al. study[8]. After reannotation with our procedures (described below), 107 exonic gDNMs from 97 trios with BD underwent our analysis (1.10 per proband). The summary of the sequence data used in this study is described in Supplementary Table 2.

**Read alignment and variant calling**. We performed read alignment and variant calling with two different pipelines for different purposes. The first pipeline (Unified pipeline) is for a comparison of gDNM rates between BD and controls in Krumm et al.[18] To this end, we performed a unified analysis using the same versions of software as in the study by Krumm et al. (BWA-mem-0.7.5a[71] and the GRCh37 reference genome for read alignment and GATK-2.7-4 for variant calling based on the GATK2 best practices workflow[72,73]). We included around 70 individuals in each batch of the joint genotyping by GATK HplotypeCaller by following the methods in the Krumm et al. study. The second pipeline (Discovery pipeline) is for extensive variant discovery and was applied to the BD exomes. In this pipeline, we used BWA-mem-0.7.17[71] and the GRCh38/hg38 reference genome for read alignment and GATK-4.0/4.1 for variant calling based on the GATK4 best practices workflow[72,73]. The joint genotyping was performed in a single batch of analysis including 257 BD trios. The detailed workflows, software versions, and purposes of the two pipelines are summarized in Supplementary Table 3. The biological relationship between the proband and the parents in 257 BD trios was confirmed by vcftools-0.1.17[74] relatedness2 (PHI score >0.2). The clinical characteristics of the study participants, DNA library information, and sequencing metrics of WES for each BD trio are summarized in Supplementary Data 7. For ASD probands and unaffected siblings (control), we used deposited VCF data ($N =$ 1772), excluding families with no variant quality score recalibration (VQSR, $N =$ 94), extremely low qualities after VQSR ($n$ of PASS variants <3000, $N =$ 17), inadequately formatted files ($N = 4$), or a suspected parent–child relationship (PHI score <0.1 calculated by vcftools-0.1.17 relatedness2, $N = 11$). We used the data from 257 BD trios and 1646 quartets with ASD probands and unaffected siblings in the downstream analysis. Read alignment and variant calling were performed blindly to the affected status of the participants.

**Detection of gDNM candidates**. We first extracted gDNM candidates from the variant calls generated by the Unified or Discovery pipeline using Triodenovo-0.06[75] with default parameters. In the Unified pipeline, we then filtered out gDNM candidates observed in the cohort of unaffected parents to exclude likely benign variants and/or sequencing artifacts as follows: for BD, we removed the variants observed in the unaffected parents of 257 BD probands (defined as the parents without diagnosis of BD, SCZAD, nor schizophrenia; $N = 503$); for controls, we split the 1646 families into 7 groups (around 240 trios/group) and removed the variants observed in the unaffected parents in each group ($N \approx 480$). We subsequently removed gDNM candidates with a DNMFilter[76] score of less than 0.90 (SNVs) or 0.95 (INDELs). We set these stringent thresholds by referring to the results of Sanger sequencing validation in pilot samples to ensure high specificity. We set a more stringent threshold for indels due to a lower validation rate when the same threshold score was applied. In the Discovery pipeline, we filtered out gDNM candidates with two or more variant alleles in the batches of unaffected parents described above ($N = 503$ for BD and $N \approx 480$ for controls). The gDNM candidates only once observed in a batch of unaffected parents were retained at this stage to detect gDNMs with high sensitivity. We then extracted gDNM candidates with a DNMFilter score of ≥0.85. We set thresholds less stringent in the Discovery pipeline than those in the Unified pipeline. We subsequently performed manual inspection with IGV-2.5.2[77] and excluded the gDNM candidates with either of the following features: (i) supported by less than two reads in IGV visualization, (ii) coinciding with other two or more variant positions in the same read (suggestive of misalignment), and (iii) with two or more reads supporting the variant in the parent(s) (likely due to transmission or systematic errors). All gDNM candidates in the Discovery pipeline were liftovered to the GRCh37 coordinate for subsequent annotations. There was no gDNM candidate that failed in liftover from GRCh38/hg38 to GRCh37.

We subjected a subset of the gDNM candidates to Sanger validation prioritizing deleterious gDNMs. The post hoc validation rates for gDNM candidates identified by the Unified and Discovery pipelines (including those detected by both pipelines) are 96.8% (92/95) and 93.4% (99/106), respectively. We considered the variants identified by these procedures as "gDNMs" and used in the downstream analyses, while some of the detected DNMs were later demonstrated to be pzDNMs as described in the "Systematic analysis of postzygotic DNMs (pzDNMs) in BD" section.

**Variant annotations**. The detected variants, including gDNMs and pzDNMs, were annotated with the following information: the effect on protein function predicted by SnpEff-4.3[78], pathogenicity inference by CADD v1.4[79], nonpsychiatric pLI scores in ExAC release 3.0[80] (nonpsych.pLI), allele frequencies in gnomAD r2.1.1 in all ($N = 125,748$) and non-neuro samples ($N = 104,068$)[27] (gnomAD.AF and gnomAD.non_neuro.AF, respectively), allele frequencies in ToMMo 3.5 K JPN[28] (ToMMo.AF), and the predicted effect of missense variants on protein function by dbNSFP 4.0a[81]. Seven algorithms for functional prediction of missense variants (SIFT[82], PolyPhen-2 HumVar and HumDiv models[83], LRT[84], MutationTaster[85], Mutation Assessor[86], and PROVEAN[87]) were adopted from previous studies[19,88,89], and the severest effect was annotated if the prediction output two or more effects (depending on the number of transcripts) for one variant. The detailed file information used in our annotation is described in Supplementary Table 4. All these items were annotated in the GRCh37 coordinates. When there are two or more gDNMs in the same gene in the same individual, we aggregated them as one gDNM with the severest annotation except for the analysis of VAF and patterns of substitutions in Supplementary Figs. 4b, c. Variant annotation was performed blindly to the affected status of the participants. DNM analyses after variant annotation (described in the subsequent sections) were not performed blindly to the affected status of the participants.

**Statistical analysis of the patterns of gDNM enrichment in BD**. We compared the gDNM rates between cases and controls by one-tailed permutation tests. For this analysis, we used the data of gDNM candidates generated by the Unified pipeline, which achieved a validation rate of 96.8% in our BD cohort. We excluded gDNM calls outside of the genomic regions defined as exons in the Krumm et al. study, on sex chromosomes, and repeat regions defined by RepeatMasker (downloaded from the UCSC Table browser: http://genome.ucsc.edu/cgi-bin/hgTables) to further ensure gDNM call accuracy and avoid confoundings. For sample-level quality control (QC), we excluded the outliers in per-individual gDNM counts deviated from the expected Poisson distributions. Outlier individuals were defined by using the $dpois$ function of R as follows: case $\alpha_i$ with gDNM count $x_i$ was considered as an outlier if dpois($x_i$, lambda = $u$)*$n$ was <0.05, where $u$ is an average gDNM count and $n$ is a total gDNM count including case $\alpha_i$. This procedure was performed iteratively from the case with the maximum gDNM count in each diagnostic group. After the sample-level QC, we included 257 BD trios and 1640 ASD/control quartets in the comparison. The mutation rates were adjusted with the overall rate of synonymous gDNMs (including those observed in gnomAD or ToMMo) in controls, assuming that the rates of synonymous gDNMs are not greatly different across case and control groups from different ethnicities, based on the results in the Iossifov et al.[17] and Howrigan et al.[15] studies.

Based on the annotation by SnpEff-4.3 and CADD Phred-scaled scores, we stratified the exonic gDNM calls into (i) LoF: nonsense, frameshift indel, and canonical splice, (ii) damaging missense/inframe indel: missense variants with a CADD score >15, missense variants at a structural (protein–protein) interaction interface, stop-lost, start-lost, and inframe indel, (iii) non-damaging missense: missense variant with a CADD score ≤15, and (iv) synonymous groups. The non-exonic DNMs (e.g., untranslated region and intronic DNMs detectable by an exome analysis) did not undergo the subsequent analysis. The threshold for the CADD score was preset according to previous studies[8,90]. We used a permutation test because we could not assume appropriate parametric distribution due to the small number of some classes of mutations (e.g., non-damaging missense). The filtering based on the allele frequencies in the general population was performed using the information of "gnomAD.non_neuro.AF" and "ToMMo.AF" annotated above. We defined evolutionarily constrained genes as genes with nonpsych.pLI >0.90 (3488 genes). The gDNMs from previous publications were reannotated with the same procedures as those for BD. The sources for previous publications are listed in Supplementary Table 1.

**GO enrichment analysis**. We performed GO enrichment analysis using DNENRICH[13] with one million random permutation, a statistical software package that considers gene length and trinucleotide contexts (e.g., high rates of C to T transitions at CpG sites), both of which are known to significantly influence per-gene mutation rates. The input deleterious gDNMs in all the genes in BD were gDNMs identified by our Unified/Discovery pipelines and gDNMs reported in the study by Goes et al.[8] (total $n$ of deleterious gDNMs = 171 in 169 unique genes). The DNM candidates that were not validated by Sanger sequencing were not included. The input deleterious gDNMs in control are compiled from ref. [20] (Supplementary Table 1). Although the analytical pipeline used in ref. [20] is not identical to that used in our study, gDNMs were detected from GATK-based variant calls in both of our and their studies. The resulting gDNM count per trio in our study (1.12) was similar to that in ref. [20] (1.16). The slightly higher gDNM rate in ref. [20] may reflect a lower proportion of gDNM calls subjected to validation experiments in their study.

The background genes were set as the 19,182 unique protein-coding genes compiled from the list of canonical transcripts defined in the SnpEff GRCh37.75

database[78]. We used the PantherGO-slim (PantherGOslim.obo) as the reference list of GO terms. Genes included in each GO term were compiled by aggregating the gene lists by the GO consortium[91], SynGO[30], and EnricherGO[92]. The source files are listed in Supplementary Table 1. We excluded the GO terms including a small number (<30) of genes, which have limited statistical power, and the terms including a very large number (>1500) of genes, which are not informative when exploring specific pathways. The gene set enrichment analysis with constrained genes was also performed using DNERNCH with the same input and background genes as those for the GO enrichment analysis.

Network visualization of the result of GO enrichment analysis was performed by using EnrichmentMap v3.2.1 plugin[93] of Cytoscape v3.7.2[94]. The GO terms not included in the latest version of the gmt file provided by the developer of EnrichmentMap (Human_GO_AllPathways_no_GO_iea_March_01_2020_symbol.gmt) were not used for the network visualization. Nodes were connected when the overlap coefficient of the containing genes >0.5. GO terms with raw $P < 0.05$ and clusters with two or less terms are not shown in the figure.

**Gene set enrichment analysis integrating transcriptome data of bulk tissues.** To perform gene set enrichment analysis integrating transcriptome data of bulk tissues, we used the GTEx (v8)[31] and the BrainSpan[32] datasets. For each of the 54 human bulk tissues in GTEx or the 60 datasets from combinations of 6 brain regions and 10 developmental periods in BrainSpan, we extracted the top 2% of the genes with the highest expression and used them as the "gene set". We then performed DNENRICH analysis to statistically test whether the genes hit by deleterious gDNMs are enriched for each gene set characteristic to a tissue or a developmental period of a brain region with one million random permutations. For the DNENRICH analysis, we used the same background gene list and the parameters as in the GO enrichment analysis. A comparison between 13 brain and 41 non-brain tissues in GTEx or between 24 fetal (six brain regions in four periods; Early Fetal, Early Mid Fetal, Late Mid Fetal, and Late Fetal) and 36 postnatal (six brain regions in six periods; Neonatal Early Infancy, Late Infancy, Early Childhood, Middle Late Childhood, Adolescence, Young Adulthood) periods in BrainSpan was performed by the exact Wilcoxon rank-sum test by sorting the datasets in the order of the uncorrected $P$ values from the DNENRICH analysis.

**Enrichment analysis integrating single-cell (nucleus) RNA sequencing data.** Single-nucleus RNA sequencing data (gene-level exonic read counts of 7283 nuclei in "human_ACC_2018-10-04_exon-matrix.csv" file) of adult human ACCs were downloaded from the Allen Brain Map Cell Types Database (https://portal.brain-map.org/atlases-and-data/rnaseq). Being informed by the violin plots of QC metrics, we first excluded cells with (i) <1000 detected genes (possible low-quality cells or empty droplets), (ii) >10,000 detected genes or 1,500,000 uniquely mapped reads (possible cell doublets), and (iii) >2% of reads mapped onto the mitochondria genome (possible low-quality or dying cells) (Supplementary Fig. 5a, b). In total, 6296 cells passed the filters. We then performed data normalization (with scale. factor = 10,000), feature selection (with nfeatures = 2000), linear transformation, principal component analysis, clustering (with resolution = 0.5), and dimensional reduction (by UMAP) following the Guided Clustering Tutorial of Seurat v3.1.2[95]. According to the result of an elbow plot, we used the first 20 principal components for clustering analysis (Supplementary Fig. 5c). Annotation of cell clusters was performed based on the expression of marker genes (e.g., *SLC17A7* for excitatory neurons, *SST* and *VIP* for subtypes of interneurons, *GFAP* for astrocytes, etc.) and/ or the results of the analysis of genes differentially expressed across clusters using the FindMarkers function of Seurat.

Cells preferentially expressing the genes hit by deleterious gDNMs in BD (169 unique genes in Supplementary Data 5) was identified by AUCell v1.8.0[35], a tool scoring cells by the area under the curve (AUC) drawn by the rank of gene expression level and the cumulative number of genes in the gene set of interest at each expression level-based rank (Supplementary Fig. 5d). To avoid potential bias due to the number of expressed genes in each cell, we used the information of the most highly expressed 1000 genes per cell (with aucMaxRank = 1000) for the AUCell analysis (as we excluded cells with <1000 expressed genes in the above described preprocessing, every cell should have >1000 expressed genes). When there are multiple genes with the same unique read count, the ranks of these genes were randomly shuffled (the default behavior of AUCell). Top 5% of the cells preferentially expressing the genes hit by deleterious gDNMs in BD (i.e., those with the highest AUC-based score) were colored in Fig. 3c. The proportion of the cells preferentially expressing the genes hit by deleterious gDNMs in BD in each cluster was compared with the theoretical expectation (i.e., 5%) by one-tailed binomial test with Bonferroni correction (16 tests). Enrichment of the c8 signature genes (genes upregulated in the cluster 8 at FDR < 0.05) among the genes hit by deleterious gDNMs in BD or controls was tested by DNENRICH as described above.

GO enrichment analysis of the c8 signature genes was performed by ToppGene[96] enabling the "find alternatives for missing symbols" option. Network visualization of the result of GO enrichment analysis was performed using the EnrichmentMap v3.2.1 plugin as described above. Significance of overlap of the c8 signature genes with known DD genes[37] or the index genes at the genomic loci genome-wide significantly associated with BD in Stahl et al.[4] was evaluated by hypergeometric tests. For this analysis, we used the intersection of the lists of

protein-coding genes[97] and genes expressed in the cluster 8 as the background (15,713 genes). The c8 signature genes, known DD genes, and the index genes at BD GWAS loci not in this list of background genes were excluded from the analysis (Supplementary Data 5).

**Statistical assessment of genes recurrently hit by deleterious gDNMs.** Statistical significance for the observed numbers of deleterious gDNMs in a gene was assessed by using an established model of per-gene mutation rates provided by Samocha et al.[39] Like DNENRICH, this model considers gene sizes and local trinucleotide contexts in the calculation of the mutation rates. We calculated an expected number of deleterious gDNMs in a gene of interest using the provided mutation rate and the size of the cohort and then compared the observed and expected numbers using one-tailed Poisson test to obtain a $P$ value, assuming that the gDNM counts in a sufficiently large population follow the Poisson distribution. The mutation rate for damaging missense gDNMs (CADD score >15) in a gene was calculated by multiplying the proportion of damaging missense gDNMs over all missense gDNMs in controls of the Satterstrom et al.[20] study and the mutation rate for all missense gDNMs provided in the Samocha et al. study. We did not include inframe indel gDNMs in this analysis because mutation rates for inframe indels are not provided in this mutational model. The exome-wide significance threshold was defined as $P = 2.74 \times 10^{-6}$ based on the number of genes with available mutation rates in Samocha et al. ($n = 18,271$).

**Detection of pzDNM candidates.** Candidates for pzDNMs were detected by Mutect2, following the GATK4 best practices workflow for somatic short variant discovery. The recalibrated bam files generated by the Discovery pipeline were used as the inputs. We analyzed Japanese BD trios ($N$ of trios = 171) for pzDNM discovery considering the availability of the DNA samples for experimental validation. We constructed a "Panel of Normals" from the parents without BD, SCZAD, nor schizophrenia ($N = 503$) to systematically exclude sequencing artifacts, misalignments, and likely benign variants frequently observed in control individuals. We also excluded the candidates hitting suspected copy-number variation (CNV) regions called by XHMM-1.0[98] to prevent false pzDNM calls due to CNVs. From the obtained list of pzDNM candidates, we selected the variants fulfilling the following criteria: (i) VAF in the proband <0.35, (ii) number of reads supporting the variant in the proband ≥5, (iii) supported by both forward and reverse reads, (iv) number of reads supporting the variant in each parent ≤1, (v) read depth in each parent ≥10, and (vi) TLOD (fidelity score by GATK-4.1.0.0 MuTect2) ≥5. The full pipeline is described in Supplementary Table 5.

**Validation of gDNM and pzDNM.** The candidates of gDNM underwent validation experiment of Sanger sequencing for the proband, the mother, and the father. All the candidates in the previous study[7] had been already validated. We selected potentially relevant gDNM candidates and candidates with low DNMFilter scores for Sanger validation.

An LoF pzDNM in *KMT2C* (p.Lys3601*) was validated by direct sequencing of the cloned PCR fragments containing the p.Lys3601* variant and a nearby heterozygous common SNP (rs74483926). The candidates for pzDNM detected by Mutect2 were subjected to TAS by MiSeq (Illumina) using v2 reagents with paired-end 151 bp reads. We designed PCR primers for candidate sites that yielded single-banded PCR amplicons of expected sizes by Primer3plus[99]. Samples with poor DNA quality were omitted in the validation experiment. Sequencing libraries for MiSeq were prepared by two rounds of PCRs and analyzed as previous studies[44,100]. The sequence reads were aligned to GRCh38/hg38 by BWA-mem-0.7.17. We then calculated the VAF by counting the number of reads supporting a variant with a base quality ≥30 and mapping quality (mapQ) ≥60 by bam-readcount (https://github.com/genome/bam-readcount) with respect to the total number of reads mapped onto the candidate site (minimum mapQ60 depth >3500, median mapQ60 depth = 11,776). We extracted the variants with a VAF ranging from 1 to 47.5% in TAS as the validated pzDNMs and used in the downstream analysis. Variants with a VAF ≥47.5% were considered as likely gDNMs. The $P$ value from a binomial test to an ALT count out of a total base count (assuming a theoretical rate = 0.5) in each validated pzDNM was almost 0 or <$4.93 \times 10^{-194}$ in the calculation by R (Supplementary Data 6). The primer sequences for the validation experiments in this study are listed in Supplementary Data 8.

**Substitution patterns in gDNM and pzDNM.** To compare the patterns of nucleotide substitutions in gDNMs ($n = 364$) and pzDNMs ($n = 45$), we classified the DNMs into the following 7 types of substitutions: A>C, A>G, A>T, C>A, C>G, C>T (CpG context), and C>T (non-CpG context). The similarity between the substitutions of gDNMs and pzDNMs was assessed by Fisher's exact test (two categories of DNM × seven types) and Pearson correlation coefficient (Supplementary Fig. 4c). Based on the observed similarity of substitution types between gDNMs and pzDNMs, we concluded that the rates of gDNMs for each gene[39] can be applied to the analysis of pzDNMs.

**Statistical assessment of DD genes hit by deleterious pzDNMs/gDNMs in BD.** Statistical significance for the observed numbers of deleterious pzDNMs/ gDNMs in 299 DD genes[37] in BD was assessed based on the per-gene mutation

rates[39] of these genes. We calculated an expected proportion of deleterious pzDNMs/gDNMs in DD genes using the sum of the deleterious mutation rates for the 299 DD genes and for all the protein-coding genes and then compared the observed and expected proportions using one-tailed binomial test to obtain a $P$ value. The mutation rate for damaging missense DNMs (CADD score >15) in a gene was calculated by multiplying the proportion of damaging missense DNMs over all missense DNMs in controls of the Satterstrom et al. study[20] and the mutation rate for all missense DNMs provided in the Samocha et al. study[39]. We did not include inframe indel DNMs in this analysis because mutation rates for inframe indels are not provided in this mutational model. As we did not have a comparable pzDNM list derived from age-matched healthy controls with the harmonized pipeline, we assessed the statistical significance of DD genes in the genes hit by deleterious pzDNMs using the theoretical expectation.

**Statistical assessment of observing genes recurrently hit by pzDNMs**. We assessed the probability of gene-level recurrence (i.e., observation of two or more probands carrying a DNM in the same gene) as an exome-wide simulation $P$ value by randomly generating 26 DNMs (the number of deleterious pzDNMs [including those observed in the general population] identified in our study) in exome one million times. Random generation of DNMs was performed by using DNENRICH taking gene length and trinucleotide contexts into account. The probability for our observation (one gene-level recurrence at *SRCAP*) was calculated by the number of permutations where one or more genes are hit by multiple DNMs. The background genes were set as the 19,182 unique protein-coding genes compiled from the list of canonical transcripts defined in the SnpEff GRCh37.75 database.

**Compilation of genes associated with CHIP**. To assess the possibility that *SRCAP* is a gene frequently mutated in clonally expanded hematopoietic cells, we compiled a list of genes associated with CHIP by aggregating the information of CHIP candidate genes[101–106] (Supplementary Table 6) and the data of pzDNMs in a previous study[107] (see Supplementary Note 1 for detail).

**Structural consideration of SRCAP**. Structural consideration and figure preparation were carried out with PyMOL (Schrodinger, New York, NY, USA). For structural comparison, the Human SRCAP and yeast SWR1 complexes were superposed using the "super" command of PyMOL. The structural model of the p. Leu696Phe substitution of the yeast SWR1 protein was created by using the "Mutagenesis" wizard of PyMOL.

**Reporting summary**. Further information on research design is available in the Nature Research Reporting Summary linked to this article.

## Data availability

De novo mutations discovered from BD trios are listed in Supplementary Data 1 and 6. The sequence data of study participants who provided informed consent for database registration ($N$ of trios = 144) are available through the National Bioscience Database Center (NBDC) Human Database, Japan with accession code JGAS000273/JGAD000379. The sequence data from the trios with no consent for database registration ($N$ of trios = 27) can be only accessed via formal collaboration due to the contents of the obtained informed consent. This study's key resources are summarized with source information in Supplementary Tables 1, 2, and 4. We used the following data from the public database: exome data from trios with schizoaffective disorder (NCBI dbGaP phs000687.v1.p1 with authorization) [https://www.ncbi.nlm.nih.gov/gap/], exome data from quartets with autism spectrum disorder (SFARI SSC WES3 with authorization) [https://www.sfari.org/resource/sfari-base/], de novo mutations in bipolar disorder [https://doi.org/10.1038/s41380-019-0611-1], GRCh37 human reference genome [ftp://gsapubftp-anonymous@ftp.broadinstitute.org/bundle/b37/], GRCh37 variant information files [ftp://gsapubftp-anonymous@ftp.broadinstitute.org/bundle/b37/], hg38 human reference genome [https://console.cloud.google.com/storage/browser/genomics-public-data/resources/broad/hg38/v0], hg38 variant information files [https://console.cloud.google.com/storage/browser/genomics-public-data/resources/broad/hg38/v0], ExAC 0.3 nonpsychiatric pLI [ftp://ftp.broadinstitute.org/pub/ExAC_release/release0.3/functional_gene_constraint/], gnomAD r2.1.1 [https://gnomad.broadinstitute.org/downloads], and ToMMo 3.5JPN [https://jmorp.megabank.tohoku.ac.jp/202001/downloads/legacy/#variant]. Source data are provided with this paper.

## Code availability

All the software packages used for the analyses are publicly available. Custom codes are available upon request.

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

## Acknowledgements

We thank all the participants of this study. We thank the Support Unit for Bio-Material Analysis at RIKEN CBS, with special thanks to K. Fukumoto, for assisting sequencing library preparation. We are grateful to Dr. Y. Kageyama for clinical assessment, all the members of the Laboratory for Molecular Dynamics of Mental Disorders at RIKEN for valuable discussion, and the RIKEN HOKUSAI team for managing a high-performance computing system. This work was partly supported by AMED under Grant Number JP20km0405208 (the Advanced Genome Research and Bioinformatics Study to Facilitate Medical Innovation [GRIFIN] for T.K.), JP20dm0207074 (the Brain/MINDS for T.K.), JP20dm0107133 (Strategic Research Program for Brain Sciences [SRPBS] for A.T.), JP20dm0307028 (Strategic International Brain Science Research Promotion Program [Brain/MINDS Beyond] for M.N. and A.T.), JP20km0405214 (Platform Program for Promotion of Genome Medicine for A.T.), and JP20ek0109381 (Practical Research Project for Rare/Intractable Diseases for A.T.) as well as JSPS KAKENHI under Grant Number JP18H05435 (T.K.), JP18H05428 (T.K.), JP20H03605 (K.M.), JP16H06254 (A.T.), JP20H05777 (A.T.), JP18K15479 (M.N.), and JP16H06277 (M.N.). The Bulgarian Trio Sequencing study is an accumulation of exome sequencing performed and/or funded by the Broad Institute, Cardiff University, Icahn School of Medicine at Mount Sinai, and the Wellcome Trust Sanger Institute. Work at the Broad Institute was funded by Fidelity Foundations, the Sylvan Herman Foundation and philanthropic gifts from Kent and Liz Dauten, Ted and Vada Stanley, and an anonymous donor to the Stanley Center for Psychiatric Research. Work at Cardiff was supported by Medical Research Council (MRC) Centre (G0800509) and Program Grants (G0801418), the European Community's Seventh Framework Programme (HEALTH-F2-2010-241909 (Project EU-GEI)). Work at the Icahn School of Medicine at Mount Sinai was supported by the Friedman Brain Institute, the Institute for Genomics and Multiscale Biology, and the National Institutes of Health grants R01HG005827 (SMP) and R01MH071681 (PS). Work at the Wellcome Trust Sanger Institute was supported by The Wellcome Trust (WT089062 and WT098051). The recruitment of the trios in Bulgaria was funded by the Janssen Research Foundation. We are grateful to all of the families at the participating Simons Simplex Collection (SSC) sites, as well as the principal investigators (A. Beaudet, R. Bernier, J. Constantino, E. Cook, E. Fombonne, D. Geschwind, R. Goin-Kochel, E. Hanson, D. Grice, A. Klin, D. Ledbetter, C. Lord, C. Martin, D. Martin, R. Maxim, J. Miles, O. Ousley, K. Pelphrey, B. Peterson, J. Piggot, C. Saulnier, M. State, W. Stone, J. Sutcliffe, C. Walsh, Z. Warren, E. Wijsman). We appreciate obtaining access to SSC_WES3 data on SFARI Base under SFARI_ID 2382.1.2.

## Author contributions

M.N., T.K., and A.T. conceived the study design. N.S., T.H., K.F., K.M., M.K., N.I., M. Ikeda, and T.K. performed clinical assessment. Y.W. and T.O. helped the enrollment of participants. A. Komori and M. Ishiwata performed DNA extraction. M.N., A. Kazuno, A. Komori, N. Matoba, and T.K. were in charge of resource management for BD samples. N. Matsumoto and A.T. were in charge of data management for SFARI data. M.N., A.N.A., K.H., and A.T. performed bioinformatics analyses. A. Kazuno and T.N. performed molecular biological experiments. M.N., T.T., T.K., and A.T. supervised molecular biological experiments. T.S. and K.O. performed structural modeling. A.T. supervised the bioinformatics and statistical analysis. M.N., T.K., and A.T. wrote the original draft. T.K. and A.T jointly directed the project and are listed in the author list in alphabetical order.

## Competing interests

The authors declare no competing interests.
