## [Peer Review File · Nature Communications]

Reviewers' Comments:

Reviewer #1:

Remarks to the Author:

In this interesting and technically well-done study, Takata et al. seek rare de novo mutations in bipolar disorder. While two previous such studies have been published, sample sizes were small and results did not establish any single gene or gene set as carrying an increased burden of DNMs in BD.

The authors extract candidate DNMs from whole-exome/genome sequencing data obtained in 354 trios, including 88 with "schizoaffective disorder." This analysis generates a list of 241 apparently novel DNMs. Genes with $pLi > 0.9$ are significantly enriched for LoF DNMs in BD. This enrichment is stronger when BDII and BD-NOS cases are excluded and remains significant when schizoaffective disorder cases are excluded. These were then combined with genes from a previous de novo study, generating a combined list of genes significantly enriched for synapse and ion channels, among other pathways.

The authors subsequently analyze scRNA-sequencing data from human anterior cingulate cortex, where they claim to identify a subset of excitatory neurons that preferentially express genes with deleterious DNMs in BD. One DNM appears to be mosaic in the carrier, leading to a further investigation of so-called pzDNMs in BD. The authors then claim to find significant enrichment of deleterious pzDNMs in known developmental disorder genes.

Authors conclude that DNMs contribute "to the genetic architecture of BD," providing "insights into its neurobiology" and suggesting the hypothesis that pzDNMs in DD genes also "cause" BD.

These results are not conclusive, but do raise interesting questions about the potential role of de novo mutation and somatic mosaicism in BD, deserving of further study. There are a few issues that if addressed would substantially improve the manuscript:

1. The gene ontology enrichment results appear to be based largely on mutations observed in high pLi genes. Any set of high pLi genes will typically be enriched for synapse, ion channels, etc., all of which are highly conserved genes. Thus it is critical to specify in the enrichment analysis a background list of all genes whose pLi distribution matches that of the input set. Similar concerns apply to the tissue and age enrichment results.
2. The putative somatic mutations reported herein are an interesting and potentially important novel finding for bipolar disorder. The argument that these constitute somatic variants in relevant tissues is weak, however. Have the authors considered sequencing of post-mortem brain tissue obtained from donors with BD? Somatic variants detected in ACC or other relevant brain regions would constitute strong evidence in support of the somatic mutation hypothesis.
3. It is not clear that the pzDNM rate estimated in the bipolar trios is any different than what would be observed in blood-derived DNA from a comparably aged comparison sample. For purposes of comparison, it looks like the authors used unaffected siblings of ASD probands. These are probably much younger than the BD probands.
4. Absent experimental studies that establish causation, authors should refrain from using words like "cause."

Reviewer #2:

Remarks to the Author:

This paper reports on an analysis of exome-wide de novo mutations in trios with bipolar disorder to test if such mutations contribute to risk of bipolar disorder. De novo mutations have been shown to play a role in intellectual disabilities, autism, and also schizophrenia. Their role in bipolar disorder is less clear and, therefore, remains an open question. This is one of the largest samples of trios to date to examine this question. It builds on earlier work by this group and also utilizes published data from

one of the only other relatively large samples of trios in bipolar disorder. It also uses existing sequencing data from autism quartets, processed here with the same de novo calling pipeline, to enable comparisons of de novo mutations rates with controls (in this case unaffected siblings in the autism families). The study finds there is a significant increased rate of loss-of-function (and to a lesser extent damaging missense) de novo mutations in evolutionary conserved genes – a pattern which has been observed in autism and schizophrenia. The fact that a similar increased rate is not observed for non-damaging missense or synonymous de novo mutations increases confidence that the finding of interest is not due to some technical bias (like the fact that they are comparing rates in bipolar and unaffected trio samples that were sequenced separately). Follow-up work by the group suggests the observed de novo mutations are enriched in genes at the presynaptic active zone, as well as in a specific subset of excitatory neurons of the sACC that are also characterized by high expression of known developmental disorder genes. A particularly novel finding of the study is that in addition the germline de novo mutations, there is evidence for increased post-zygotic mutations in bipolar disorder and that these may overlap more than expected by chance in known developmental disorder genes. This raises the interesting hypothesis that post-zygotic mutations of certain developmental disorder genes may contribute to bipolar disorder risk. Overall, this is an important and well done study that is clearly reported. There is a great deal of interest to determine whether de novo mutations contribute to bipolar disorder risk as they seem to do with the other severe developmental disorders. This study provides the largest sample to date to look at this question and provides compelling evidence that indeed they do. It also raises an interesting hypothesis about the role of post-zygotic mutations in bipolar disorder that based on the findings reported here merit further investigations. The methods used in the study are state of the art and well executed. I have only a couple minor comments or questions for clarification.

- 1) Can the investigators clarify if the 88 schizoaffective probands included in the study are bipolar or depressive sub-type?
- 2) Can the investigators clarify if the current sample includes and builds on the sample first reported by Kataoka et al. in *Molecular Psychiatry* in 2016? I may have missed it, but it would be helpful to state this explicitly in the text so readers will have the appropriate context of how this sample fits with previous data that has been reported.
- 3) Supplementary Table 5 that was provided has the title “Genesets used in cell type enrichment analysis”, but the contents of the table provides information about the genes hit by de novo mutations, C8 signature genes, genes in GWAS loci, and their overlap – so it looks like the title is mis-labeled?
- 4) The “g” label is missing Fig 4.

Point-by-point response

We would like to thank the Reviewers for their encouraging and constructive comments. Below is our point-by-point response to the specific remarks of each Reviewer. We have revised the manuscript including the responses to the Reviewers' comments. The comments of the reviewers are in *blue italic*, and the revised parts of the manuscript are in *red*.

Reviewer #1

In this interesting and technically well-done study, Takata et al. seek rare de novo mutations in bipolar disorder. While two previous such studies have been published, sample sizes were small and results did not establish any single gene or gene set as carrying an increased burden of DNMs in BD. The authors extract candidate DNMs from whole-exome/genome sequencing data obtained in 354 trios, including 88 with "schizoaffective disorder." This analysis generates a list of 241 apparently novel DNMs. Genes with $pLi > 0.9$ are significantly enriched for LoF DNMs in BD. This enrichment is stronger when BDII and BD-NOS cases are excluded and remains significant when schizoaffective disorder cases are excluded. These were then combined with genes from a previous de novo study, generating a combined list of genes significantly enriched for synapse and ion channels, among other pathways. The authors subsequently analyze scRNA-sequencing data from human anterior cingulate cortex, where they claim to identify a subset of excitatory neurons that preferentially express genes with deleterious DNMs in BD. One DNM appears to be mosaic in the carrier, leading to a further investigation of so-called pzDNMs in BD. The authors then claim to find significant enrichment of deleterious pzDNMs in known developmental disorder genes. Authors conclude that DNMs contribute "to the genetic architecture of BD," providing "insights into its neurobiology" and suggesting the hypothesis that pzDNMs in DD genes also "cause" BD. These results are not conclusive, but do raise interesting questions about the potential role of de novo mutation and somatic mosaicism in BD, deserving of further study. There are a few issues that if addressed would substantially improve the manuscript:

We would like to thank the reviewer for the succinct summarization of our results and valuable comments. We appreciate the critical issues raised by the reviewer. We have revised the following four points, clarifying several confusing points in the original manuscript.

1. The gene ontology enrichment results appear to be based largely on mutations observed in high pLi genes. Any set of high pLi genes will typically be enriched for synapse, ion channels, etc., all of which are highly conserved genes. Thus it is critical to specify in the enrichment analysis a background list of all genes whose pLi distribution matches that of the input set. Similar concerns apply to the tissue and age enrichment results.

We apologize if there was a confusing description. First, we would like to clarify that we used all the deleterious germline de novo mutations (gDNMs) in bipolar disorder, not the deleterious gDNMs in high pLI genes, as the input of the gene ontology and tissue and the age enrichment analyses. As there is no a priori selection of deleterious gDNMs in high pLI genes in our analyses, we used a standard set of all protein-coding genes as the background (canonical coding genes defined by SnfEff-4.3).

While the genes hit by deleterious gDNMs in bipolar disorder are actually enriched for high pLI genes ($P = 0.00549$, gene set enrichment analysis by DNENRICH [Fromer et al., Nature 2014, ref. 13]), this is a result of an unbiased comprehensive analysis without any preselection. Therefore, we believe that our analysis, which uses all deleterious gDNMs as the input and all protein-coding genes as the background, is justifiable. We would also like to note that similar analyses including gene ontology analysis were performed in studies of autism spectrum disorders and schizophrenia, in which it is well known that deleterious gDNMs are enriched in high pLI gene (Fromer et al., Nature 2014, ref. 13 and Satterstrom et al., Cell 2020, ref. 20).

In addition, the same analytical procedure was applied to the deleterious gDNMs in control and no enrichment for synapse/ion-channel genes was observed in the genes hit by all the deleterious gDNMs in control (**Fig. 2a**). Therefore, we would like to argue that the enrichment of synaptic/ion-channel genes in bipolar disorder is not derived from an analytical artifact.

Meanwhile, we agree that high pLI genes are enriched for synaptic and ion-channel genes as shown in the Koopmans et al. study (Neuron 2019, ref. 91). Therefore, it would be unsurprising to observe an enrichment of such gene sets in genes hit by deleterious gDNMs in bipolar disorder. We clarified this point as follows.

Page 7, Line 169 in the revised manuscript

The genes hit by deleterious gDNMs in BD are enriched for constrained genes ($P = 0.00549$, DNENRICH analysis), while constrained genes are known to be enriched in synaptic genes⁹¹. Thus, the enrichment of synaptic genes including ion-channel genes in the genes hit by deleterious gDNMs in BD would be reasonable.

Accordingly, the analytical procedure was added in the Methods section.

Page 32, Line 825 in the revised manuscript

The gene set enrichment analysis with constrained genes was also performed using DNERNCH with the same input and background genes as those for the GO enrichment analysis.

2. The putative somatic mutations reported herein are an interesting and potentially important novel finding for bipolar disorder. The argument that these constitute somatic variants in relevant tissues is weak, however. Have the authors considered sequencing of post-mortem brain tissue obtained from donors with BD? Somatic variants detected in ACC or other relevant brain regions would constitute strong evidence in support of the somatic mutation hypothesis.

We agree with the reviewer that the somatic variants in the brain tissues are important targets to elucidate the contribution of somatic variants to bipolar disorder. We are now considering sequencing of post-mortem brain tissues obtained from donors with bipolar disorder. In our continuous studies on somatic variants in psychiatric disorders, however, we would like to say that investigation of the brain samples is beyond the scope of our current study. To the best of our knowledge, there is no report of a comprehensive analysis of somatic variants (pzDNMs in our manuscript) in the peripheral tissues in bipolar disorder. Therefore, we believe that our current study is worth reporting *per se*. We regard the investigation of somatic variants in the peripheral and brain tissues as closely related but separate projects. Our current study using the peripheral tissues is the first step to investigate the association of somatic variants and bipolar disorder. We are to proceed to the investigation of somatic variants in the human brain as the next step, after collecting a sufficient number of brain samples. We have added a discussion about this issue as follows.

Page 20, Line 515 in the revised manuscript

Investigation with additional samples including postmortem brain tissues derived from donors with bipolar disorder will further advance our understanding of the contribution of pzDNMs.

3. It is not clear that the pzDNM rate estimated in the bipolar trios is any different than what would be observed in blood-derived DNA from a comparably aged comparison sample. For purposes of comparison, it looks like the authors used unaffected siblings of ASD probands. These are probably much younger than the BD probands.

We thank this reviewer for pointing out this very important issue. We would like to first clarify that we did not use the data of pzDNMs in the unaffected siblings of ASD probands as the control.

We agree with the concern of this reviewer about the age difference when analyzing pzDNMs. Age-related clonal hematopoiesis of indeterminate potential (CHIP) would influence the pzDNM rates in our recruited samples and the unaffected siblings of ASD probands. This is exactly why we refrained from using the list of pzDNMs in the unaffected siblings of ASD probands in our study. Also, we considered that it is inappropriate to use the data of pzDNMs in the parents in the ASD quartets (Krupp et al., American Journal of Human Genetics 2017, ref. 107) as the referential control for the following two reasons: (i) some of the parents with ASD child have pzDNMs in ASD candidate genes, which are transmitted and detected as germline de novo mutations in the ASD probands, as reported in the Krupp et al. study (ref. 107), and (ii) the pipeline of pzDNM analysis for the parents in ASD quartets are different from the pipeline used in pzDNM analysis for the probands with bipolar disorder, which could cause a significant bias in the detection of pzDNMs. Therefore, we used the theoretical mutation rates provided in the Samocha et al. study (ref. 38) after confirming the validity of applying the mutation rates in ref. 38 to pzDNM analysis (**Supplementary Fig. 4c**).

We have clarified that we did not compare pzDNM rates between bipolar disorder and control in our current study as follows.

Page 38, Line 980 in the revised manuscript

As we did not have a comparable pzDNM list derived from age-matched healthy controls with the harmonized pipeline, we assessed the statistical significance of DD genes in the genes hit by deleterious pzDNMs using the theoretical expectation.

4. Absent experimental studies that establish causation, authors should refrain from using words like "cause."

As the reviewer points out, the word "cause" is not appropriate for our current study. Our study is purely observational. We have revised the original manuscript as follows, indicating the changes as bold.

REVISION 4.1

BEFORE: Page 2, Line 52 in the original manuscript

We find significant enrichment of deleterious pzDNMs in known DD genes ($P = 0.00135$), offering a tempting hypothesis that pzDNMs of DD genes may **cause** BD.

AFTER: Page 2, Line 52 in the revised manuscript

We find significant enrichment of deleterious pzDNMs in known DD genes ($P = 0.00135$), offering a tempting hypothesis that pzDNMs of DD genes may **contribute to** BD.

REVISION 4.2

BEFORE: Page 12, Line 292 in the original manuscript

Being inspired by this observation, we systematically investigated pzDNMs in BD, primarily hypothesizing that pzDNMs of DD genes may **cause** neuropsychiatric disorders such as BD by affecting specific subsets of brain cells.

AFTER: Page 12, Line 296 in the revised manuscript

Being inspired by this observation, we systematically investigated pzDNMs in BD, primarily hypothesizing that pzDNMs of DD genes may **contribute to** neuropsychiatric disorders such as BD by affecting specific subsets of brain cells.

REVISION 4.3

BEFORE: Page 13, Line 310 in the original manuscript

When compared with this theoretical expectation, the observed proportion (0.25; four out of the 16 deleterious pzDNMs hit the DD genes) is highly unlikely to be a chance finding (Fig. 4c, left, $P = 0.00135$, one-tailed binomial test), supporting our hypothesis that deleterious pzDNMs of DD genes **may cause** BD.

AFTER: Page 13, Line 314 in the revised manuscript

When compared with this theoretical expectation, the observed proportion (0.25; four

out of the 16 deleterious pzDNMs hit the DD genes) is highly unlikely to be a chance finding (Fig. 4c, left, $P = 0.00135$, one-tailed binomial test), supporting our hypothesis that deleterious pzDNMs of DD genes **are associated with** BD.

REVISION 4.4

BEFORE: Page 17, Line 442 in the original manuscript

These individual examples further support a hypothesis that deleterious pzDNMs of DD genes would **cause** milder neuropsychiatric phenotypes by affecting specific subsets of brain cells.

AFTER: Page 18, Line 446 in the revised manuscript

These individual examples further support a hypothesis that deleterious pzDNMs of DD genes would **contribute to** milder neuropsychiatric phenotypes by affecting specific subsets of brain cells.

Reviewer #2

This paper reports on an analysis of exome-wide de novo mutations in trios with bipolar disorder to test if such mutations contribute to risk of bipolar disorder. De novo mutations have been shown to play a role in intellectual disabilities, autism, and also schizophrenia. Their role in bipolar disorder is less clear and, therefore, remains an open question. This is one of the largest samples of trios to date to examine this question. It builds on earlier work by this group and also utilizes published data from one of the only other relatively large samples of trios in bipolar disorder. It also uses existing sequencing data from autism quartets, processed here with the same de novo calling pipeline, to enable comparisons of de novo mutations rates with controls (in this case unaffected siblings in the autism families). The study finds there is a significant increased rate of loss-of-function (and to a lesser extent damaging missense) de novo mutations in evolutionary conserved genes — a pattern which has been observed in autism and schizophrenia. The fact that a similar increased rate is not observed for non-damaging missense or synonymous de novo mutations increases confidence that the finding of interest is not due to some technical bias (like the fact that they are comparing rates in bipolar and unaffected trio samples that were sequenced separately). Follow-up work by the group suggests the observed de novo mutations are enriched in genes at the presynaptic active zone, as well as in a specific subset of excitatory neurons of the sACC that are also characterized by high expression of known developmental disorder genes. A particularly novel finding of the study is that in addition the germline de novo mutations, there is evidence for increased post-zygotic mutations in bipolar disorder and that these may overlap more than expected by chance in known developmental disorder genes. This raises the interesting hypothesis that post-zygotic mutations of certain developmental disorder genes may contribute to bipolar disorder risk. Overall, this is an important and well done study that is clearly reported. There is a great deal of interest to determine whether de novo mutations contribute to bipolar disorder risk as they seem to do with the other severe developmental disorders. This study provides the largest sample to date

to look at this question and provides compelling evidence that indeed they do. It also raises an interesting hypothesis about the role of post-zygotic mutations in bipolar disorder that based on the findings reported here merit further investigations. The methods used in the study are state of the art and well executed. I have only a couple minor comments or questions for clarification.

We would like to thank this reviewer for appreciating our study and encouraging further investigation. We have revised our manuscript, figure, and tables, following the four remarks.

1) Can the investigators clarify if the 88 schizoaffective probands included in the study are bipolar or depressive sub-type?

Among the 88 schizoaffective probands, unfortunately, we could not specify the sub-types of 86 probands derived from the Fromer et al. study with the available data sources (Supplementary Information in the Fromer et al. study and the phenotypic information in dbGaP). We have noted this in Table S10 by introducing the following changes.

Table S10 (excel format) in the revised submission

Cells D175 - D260: SCZAD -> SCZAD*

Cell A261: [blank] -> * unknown subtype

For the other two probands in our cohort, we could clarify the sub-types. We have added the information as follows.

Table S10 (excel format) in the revised submission

Cell D138: SCZAD -> SCZAD, depressive type

Cell D155: SCZAD -> SCZAD, bipolar type

2) Can the investigators clarify if the current sample includes and builds on the sample first reported by Kataoka et al. in Molecular Psychiatry in 2016? I may have missed it, but it would be helpful to state this explicitly in the text so readers will have the appropriate context of how this sample fits with previous data that has been reported.

We are sorry for not making this point clear enough in the previous version of our manuscript. Our current study includes all the samples reported by the Kataoka et al. study in 2016. We have clarified this point in the Methods as follows.

Page 26, Line 659 in the revised manuscript

All the samples reported in our previous study⁷ were included in this study.

(ref. 7 is Kataoka et al. in Molecular Psychiatry in 2016)

3) Supplementary Table 5 that was provided has the title "Genesets used in cell type enrichment analysis", but the contents of the table provides information about the genes

hit by de novo mutations, C8 signature genes, genes in GWAS loci, and their overlap? so it looks like the title is mis-labeled?

We agree that the title of Supplementary Table 5 is not appropriate. We have changed the title of Supplementary Table 5 as follows.

Table S5 (excel format) in the revised submission

BEFORE: Cell A1

Genesets used in cell type enrichment analysis

AFTER: Cell A1

Genesets used in the characterization of c8 signature genes

4) The "g" label is missing Fig 4.

We appreciate the reviewer for his/her careful reading. We have added the label "g" in Fig. 4 as follows.

Fig. 4 (PDF format) in the revised submission

Reviewers' Comments:

Reviewer #1:

None